# Structure of the error-prone DNA ligase of African swine fever virus identifies critical active site residues

Yiqing Chen[1], Hehua Liu[1,2], Chun Yang[1], Yanqing Gao[1], Xiang Yu[1,2], Xi Chen[1], Ruixue Cui[2], Lina Zheng[2], Suhua Li[1], Xuhang Li[2], Jinbiao Ma ⬤ [2], Zhen Huang[3,4], Jixi Li ⬤ [1,5] & Jianhua Gan[1]

African swine fever virus (ASFV) is contagious and can cause highly lethal disease in pigs. ASFV DNA ligase (AsfvLIG) is one of the most error-prone ligases identified to date; it catalyzes DNA joining reaction during DNA repair process of ASFV and plays important roles in mutagenesis of the viral genome. Here, we report four AsfvLIG:DNA complex structures and demonstrate that AsfvLIG has a unique N-terminal domain (NTD) that plays critical roles in substrate binding and catalytic complex assembly. In combination with mutagenesis, in vitro binding and catalytic assays, our study reveals that four unique active site residues (Asn153 and Leu211 of the AD domain; Leu402 and Gln403 of the OB domain) are crucial for the catalytic efficiency of AsfvLIG. These unique structural features can serve as potential targets for small molecule design, which could impair genome repair in ASFV and help combat this virus in the future.

---

[1] State Key Laboratory of Genetic Engineering, Collaborative Innovation Center of Genetics and Development, Department of Physiology and Biophysics, School of Life Sciences, Fudan University, 200433 Shanghai, China. [2] State Key Laboratory of Genetic Engineering, Collaborative Innovation Center of Genetics and Development, Department of Biochemistry, School of Life Sciences, Fudan University, 200433 Shanghai, China. [3] College of Life Sciences, Sichuan University, 610041 Chengdu, China. [4] Department of Chemistry, Georgia State University, Atlanta, GA 30303, USA. [5] Department of Neurology, Huashan Hospital, Fudan University, 200040 Shanghai, China. Correspondence and requests for materials should be addressed to Z.H. (email: huang@gsu.edu) or to J.L. (email: lijixi@fudan.edu.cn) or to J.G. (email: ganjhh@fudan.edu.cn)

African swine fever virus (ASFV) is a large encapsulated double-stranded DNA virus. It belongs to the *Asfivirus* genus and is the only member of the *Asfarviridae* family. ASFV is highly contagious and can cause lethal disease in both domestic pigs and wild boars[1]. The disease caused by ASFV was first reported in Kenya in the 1920s and remained restricted to Africa till the mid 1950 s[2]. Since then, ASFV has spread into many countries in Europe, South America, the Caribbean region, as well as in Asia, especially Russian[3]. Very recently, the virus has been found in China[4], the largest pork producer in the world. The virus is turning into a global threat; however, very unfortunately, no vaccine or other useful treatment against this virus has been developed[5]. The disease has caused very serious economic problems in many countries[6]. In 2011, more than 300,000 pigs in the Russian Federation region were killed by the disease with an estimated cost of 240 million US dollars.

The size of ASFV genome varies between 170 and 190 kb, and encodes more than 150 proteins that are involved in various stages of the ASFV life cycle, including gene expression, DNA replication, virion assembly, entry into host cells, and suppression of host immune response[7]. Though the DNA synthesis process starts in the nucleus, the replication and virion assembly of ASFV are completed in the cytoplasm of infected cells[8], primarily swine macrophage cells[9]. Macrophages are very rich in free oxygen radicals[10,11], which cause constant damages to the virus genome, such as strand breaks and spontaneous depurination/depyrimidation. To efficiently overcome these DNA damages, ASFV virus has evolved its own repair system. Interestingly, unlike in humans and many other species, the fidelities of the repair DNA polymerase (*Asfv*PolX)[12] and ligase (*Asfv*LIG)[13] of ASFV are very low; they can tolerate various base mismatches at the repair sites. Therefore, in addition to the maintenance of genome stability, these repair enzymes also play important roles in strategic mutagenesis and genotype formation of ASFV. By sequencing the C-terminal region of only one gene (B6464L), more than 20 ASFV genotypes have been identified[14–16]. As demonstrated in eastern and southern Africa, existence of various genotypes further complicates the epidemiology, diagnosis, and prevention of ASFV.

Surprisingly, though ASFV virus has been extensively studied over the past 90 years, the enzymes involved in the DNA repair pathway have not been well characterized;[17,18] the structural information of ASFV repair enzymes has only begun to emerge in recent years[19–21]. The only crystal structures of *Asfv*PolX/substrate complex were reported by our group in 2017; these structures revealed a unique 5'-P binding pocket located at the finger domain of *Asfv*PolX[22]. *Asfv*LIG (Fig. 1a) is composed of 419 amino acids and is one of the smallest DNA ligases identified to date. *Asfv*LIG contains one adenylation domain (AD) in the center and one OB-fold domain (OB) at the C-terminus; both AD and OB domains are conserved in homologous proteins (Supplementary Fig. 1), including human DNA ligases (*Hs*Lig1-4)[23–25], *Sus scrofa* DNA ligase 1 (*Sus*LIG1), and archaeal ligases[26,27]. In addition to DNA repair, some DNA ligases also play a role in the ligation of Okazaki fragments occurring during replication[28]. Besides the AD and OB domains, many DNA ligases share one DNA-binding domain (DBD), which is critical for substrate binding and ligation processes. Interestingly, the N-

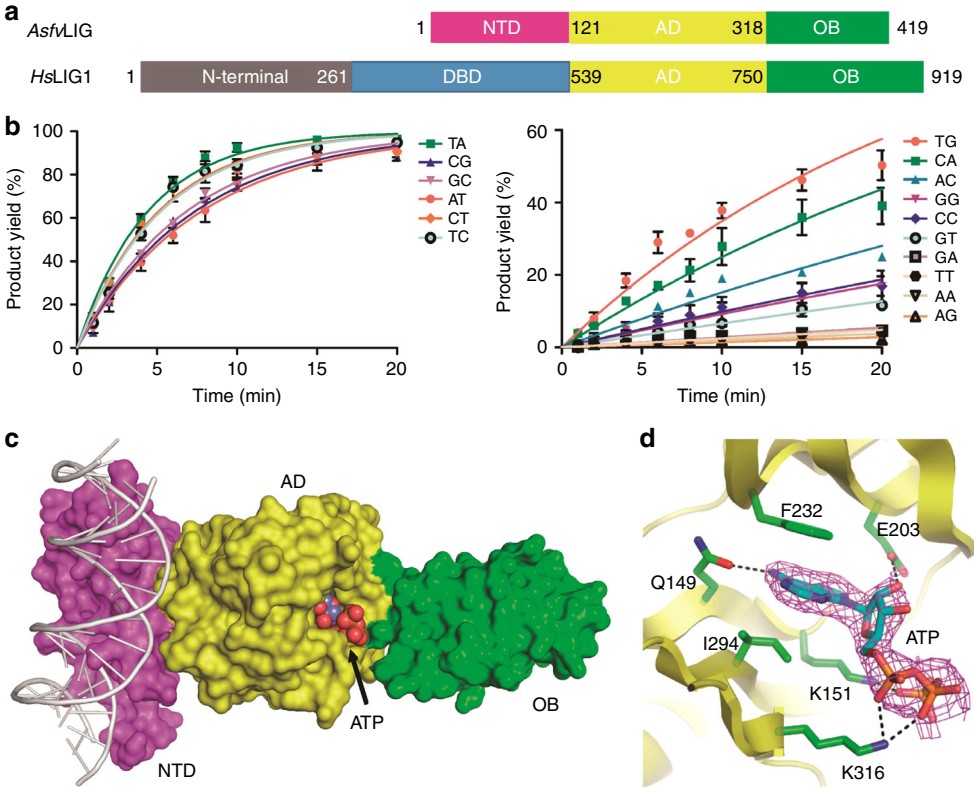

**Fig. 1** Domain architecture and ATP recognition pattern of *Asfv*LIG. **a** Schematic view comparing the domain compositions of *Asfv*LIG and *Hs*LIG1. **b** In vitro DNA ligation catalyzed by wild-type *Asfv*LIG. Data represent the mean of three independent experiments, the standard deviation (SD) values are indicated by error bars. **c** Overall fold of the non-catalytic *Asfv*LIG:DNA complex. *Asfv*LIG is shown as surface with the NTD, AD, and OB domains colored in magenta, yellow, and green, respectively. DNA and ATP are shown as cartoons and spheres in white and atomic colors (C, light-blue; N, blue; O, red; P, orange), respectively. **d** Cartoon-and-stick view showing ATP recognition by the *Asfv*LIG AD domain. ATP is shown as sticks and outlined with simulated annealing $2F_o\text{-}F_c$ omit map, which is contoured at the 1.5 sigma level

terminal region (referred to as NTD hereafter, Supplementary Fig. 1a) of *Asfv*LIG has no similarity with any other ligase DBD domains. Owing to the lack of structural information, many fundamental questions, such as overall folding, substrate recognition, and the basis for mismatch tolerance by *Asfv*LIG, remain unanswered.

Here we report the structural and functional studies of *Asfv*-LIG. Four *Asfv*LIG crystal structures, including one in the non-catalytic form and three in the catalytic form, were determined. One unsealed C:G pair, one unsealed C:T pair, and one sealed C:T pair were, respectively, captured at the catalytic sites of the three catalytic form structures, referred to as *Asfv*LIG:CG, *Asfv*LIG:CT1, and *Asfv*LIG:CT2 hereafter. Though the molecular basis underlying the low fidelity of *Asfv*LIG remains elusive, our structural and biochemical studies clearly showed that both NTD domain and the unique active site residues of *Asfv*LIG play very important roles in substrate binding and catalysis. Similar to the novel 5′-P binding pocket of *Asfv*PolX[22], the unique *Asfv*LIG NTD domain can also serve as a potential target for drug design to disrupt the DNA repair process of the ASFV genome.

## Results

**Low fidelity and ATP binding of *Asfv*LIG.** To further verify the activity and fidelity of *Asfv*LIG, we carried out in vitro catalysis using wild type (WT) *Asfv*LIG and substrates containing either Watson–Crick paired or mismatched base pairs at the ligation sites (Fig. 1b and Supplementary Fig. 2). The substrates are named as DNA-XY (Supplementary Table 1), where X and Y denote the nucleotides at the template strand and at the 3′-end upstream of the nick site, respectively. We only calculated the simplified apparent rate constants ($K_{obs}$) in this study, the full enzymatic characterization leading to both $K_{cat}$ and $K_d$ values of *Asfv*LIG has been reported by Lamarche and co-worker previously[13]. *Asfv*LIG can efficiently catalyze the ligation reactions of the substrates with Watson–Crick base pairs; though not as efficiently as previously reported[13], the ligating rate of DNA-CT is comparable to those of the Watson–Crick paired substrates (Supplementary Table 2). Interestingly, *Asfv*LIG can also efficiently catalyze the ligation reactions of several other substrates, such as DNA-TC, DNA-TG, and DNA-CA, which all possess a pyrimidine nucleotide on the template strands. Unlike the slow reaction previously reported[13], we found that the catalytic efficiency of DNA-TC is comparable to the Watson–Crick paired substrates under our reaction conditions.

To clarify the functions of the individual domains and to reveal the basis for mismatch tolerance, we crystallized the *Asfv*LIG proteins with various combinations. Though no *Asfv*LIG structure in the absence of DNA was obtained, we successfully solved four *Asfv*LIG:DNA complex structures, including one in the non-catalytic form and three in the catalytic form. Crystals of the non-catalytic form structure were grown in the presence of ATP but without Mg$^{2+}$, which is an essential cofactor of the adenylation reaction. We found that the non-catalytic structure adopts an extended fold (Fig. 1c) and the ATP is bound by the AD domain and stabilized by various interactions (Fig. 1d). Besides hydrophobic stacking with the side chains of Ile294 and Phe232, the nucleobase of ATP forms one hydrogen bond (H-bond) with Gln149, via the N6 and OE1 atoms. The sugar pucker 2′-OH group and the oxygen atoms of the phosphate groups form H-bonds with the side chains of Lys151, Glu203, and Lys316.

**Overall folding of the catalytic *Asfv*LIG:DNA complex.** Among the three catalytic structures, *Asfv*LIG:CT2 captured one sealed C:T pair, whereas *Asfv*LIG:CG and *Asfv*LIG:CT1 captured unsealed C:G and C:T pairs at the catalytic sites, respectively. The

structures belong to two different space groups: P2$_1$2$_1$2$_1$ for *Asfv*LIG:CT2 and P2$_1$ for both *Asfv*LIG:CG and *Asfv*LIG:CT1. The molecular packing of *Asfv*LIG:CT2 is very different from the other two structures in the crystal lattice. However, as indicated by the low root mean square deviation values (rmsd, about 0.75 Å based on the superposition of 404 pairs of Cα atoms) among them, the three complex structures are very similar to each other.

Owing to higher resolution (2.35 Å), the *Asfv*LIG:CT1 structure was used to demonstrate the overall folding of the catalytic form complexes (Fig. 2a and Supplementary Fig. 3). Unlike the extended non-catalytic structure (Fig. 1c), *Asfv*LIG undergoes large domain rearrangement and adopts a closed conformation in the catalytic *Asfv*LIG:DNA complex. *Asfv*LIG is assembled like a ring-shaped clamp and encircles the DNA duplex in its central channel. In the DNA, 19 out of 22 base pairs (bp) is covered by *Asfv*LIG. The AD domain of *Asfv*LIG mainly recognizes the broken strand at the nick site and the flanking region upstream, whereas the OB domain primarily interacts with the continuous template strand surrounding and downstream of the nick (Supplementary Fig. 4). In the non-catalytic form structure, the substrate DNA was bound by the NTD domains of two *Asfv*LIG molecules and adopts regular B-form conformation (Supplementary Fig. 5a); though the DNA adopts a regular B-form conformation at the two ends, it was significantly bent and unwound around the nick site in the catalytic form complex (Supplementary Fig. 5b). Similar conformational changes of the DNA have previously been observed in the *Hs*LIG1:DNA complex structure[23]. As depicted in Fig. 2b and Supplementary Fig. 5c, the relative orientations of the DNA, AD, and OB domains are similar in the two structures.

**Unique fold of *Asfv*LIG NTD and its interaction with DNA.** The NTD domain (residues 1–120) of *Asfv*LIG mimics the DBD domain (residues 262–538) of *Hs*LIG1 in the catalytic complexes (Fig. 2b and Supplementary Fig. 5c). However, unlike the AD and OB domains, which share certain sequence similarities between the two proteins (Supplementary Figs. 1b, c), *Asfv*LIG NTD has no sequence similarity to the DBDs of *Hs*LIG1 nor with any other DNA ligases. Interestingly, the overall folds of *Asfv*LIG NTD and *Hs*LIG1 DBD are significantly different from each other. In fact, as revealed by the Dali search program, the overall fold of *Asfv*-LIG NTD is completely novel. *Asfv*LIG NTD has a mixed α/β fold in nature (Fig. 3a and Supplementary Figs. 6a, b), which is composed of three α-helices (α1–α3) and seven β-strands (β1–β7). The β-strands form one flat β-sheet and α3 (residues 95–113) is packed against the β-sheet on one side.

As revealed by the *Asfv*LIG:CT1 structure, the NTD domain forms extensive interactions with the DNA substrate. The β3-β4 connecting loop (residues 23–27) points toward one end of the DNA duplex, forming three H-bonds (Fig. 3b). Lys27 forms one H-bond with the broken strand (2.9 Å, between its side chain Nz atom and the OP2 atom of G21) and the other two H-bonds are all formed by Lys24, including one (3.1 Å) between its Nz atom and the OP2 atom of template T5, and another (2.6 Å) between its main chain N atom and the OP1 atom of template C6. Unlike T5 and C6, both the OP1 and OP2 oxygen atoms of template G8 are involved in direct H-bond interactions with *Asfv*LIG NTD (Fig. 3c). OP1 interacts with the OG1 atom of Thr64 (of the β5 strand) and OP2 interacts with the NE1 atom of Trp31 (of the β4 strands).

α3 is the longest α-helix of *Asfv*LIG NTD; it lies along the major groove of the DNA duplex (Fig. 3a). α3 forms two H-bonds with the broken strand of the DNA: one (3.0 Å) between the NE2 atom of Gln98 and the OP1 atom of T8 and the other (2.9 Å) between the OG atom of Ser105 and the OP2 atom of G9

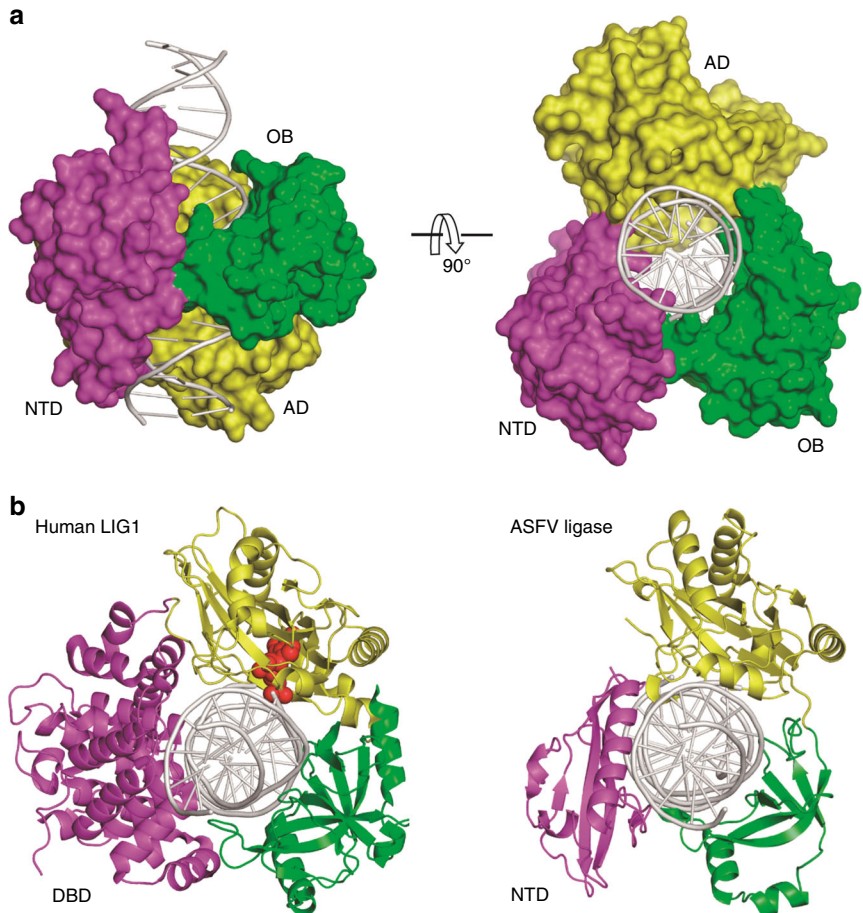

**Fig. 2** The catalytic *Asfv*LIG:DNA complex structures. **a** The overall structure of *Asfv*LIG:CT1 complex demonstrating the arrangement of individual domains in the catalytic form complex. *Asfv*LIG and DNA are shown as surface and cartoon, respectively. **b** Comparison of the catalytic form complexes of *Asfv*LIG and *Hs*LIG1 (PDB_ID: 1 × 9 N). DNAs are colored in white, whereas the ligase NTD (or DBD), AD and OB domains are colored in magenta, yellow and green, respectively. In the *Hs*LIG1 structure, the AMP that pyrophosphate linked to the 5'-P downstream of the DNA is shown as red spheres

(Fig. 3d). T8 and G9 are located in the middle of the DNA duplex. Via direct or water-mediated H-bonding (Fig. 3e), the neighboring G6 and A7 of the upstream DNA interact with Lys85 and Asn86 of the β7-α3 connecting loop (residues 82-94); the H-bond distances are all within the range of 2.7–3.1 Å. The β7-α3 connecting loop points toward the end of the DNA duplex but at the opposite direction of the β3-β4 connecting loop (Fig. 3a). Though they do not form direct H-bonds, the shape of the three tip residues (Lys89, Lys90, and Asn91) matches with the major groove and the sugar puckers of the DNA template strand (Fig. 3f). As revealed by the DNAProDB program[29], several other *Asfv*LIG NTD residues also interact with the DNA nucleotides (Fig. 3g).

**NTD is critical for catalytic complex assembly and catalysis.** In the catalytic *Asfv*LIG:DNA complex structures, the β6-β7 connecting linker of *Asfv*LIG NTD bends toward one short α-turn of the OB domain (Fig. 4a). Different from the *Hs*LIG1:DNA structure that forms several direct H-bonds between the DBD and OB domains, the NTD and OB domains of *Asfv*LIG interact with each other via two water-mediated H-bonds. Compared to the OB domain, the AD domain forms many more interactions with the *Asfv*LIG NTD domain. As depicted in Fig. 4b, one α-turn of the AD domain resides next to the C-terminus of strand β3 and helix α3 of the NTD domain. The side chain of the α-turn residue Asn307 forms two H-bonds: one (2.9 Å) between its ND2 atom and the main chain O atom of Glu22 and the other (2.8 Å)

between its OD1 atom and the side chain Nz atom of Lys114. Arg115 of α3 helix projects toward Tyr306, forming stacking interactions between their side chains (Fig. 4c). Side chains of two other residues of the NTD α3 helix are also involved in the interaction with the AD domain (Fig. 4d). The NH1 atom of Arg112 forms one H-bond (3.0 Å) with the side chain OG1 atom of Thr173; the NE2 atom of Gln113 also forms one H-bond (2.8 Å) with Thr173 but with the main chain O atom. Thr173 is located at the tip of the one loop, residing next to the nick site of the broken DNA strand.

Though *Asfv*LIG NTD has no sequence or structural similarity to *Hs*LIG1 DBD, it mimics *Hs*LIG1 DBD in the catalytic complexes (Fig. 2b and Supplementary Fig. 5c). The total numbers of DNA base pairs covered by NTD (19 bp) and DBD (18 bp) are similar in the two structures. DBD is important for DNA binding and catalysis of *Hs*LIG1: deletion of DBD lowers the substrate binding affinity by > 75-fold and reduces the catalytic efficiency of *Hs*LIG1 by > 4 × 10⁵ fold[23]. To clarify the function of *Asfv*LIG NTD, we carried out an in vitro DNA-binding assay using nick and duplex DNA-CG. WT *Asfv*LIG can bind both nick and duplex DNA-CG (Fig. 4e and Supplementary Fig. 6c); within the concentration range of 0.2–0.8 μM, the nick DNA-binding affinity of WT *Asfv*LIG is slightly higher than that of duplex DNA. Compared to WT *Asfv*LIG, the DNA-binding affinity of NTD is much weaker. At a 1.6 μM concentration, NTD also showed weak preference for the nick DNAs. Similar nick DNA preference was also observed for *Hs*LIG1 and its DBD

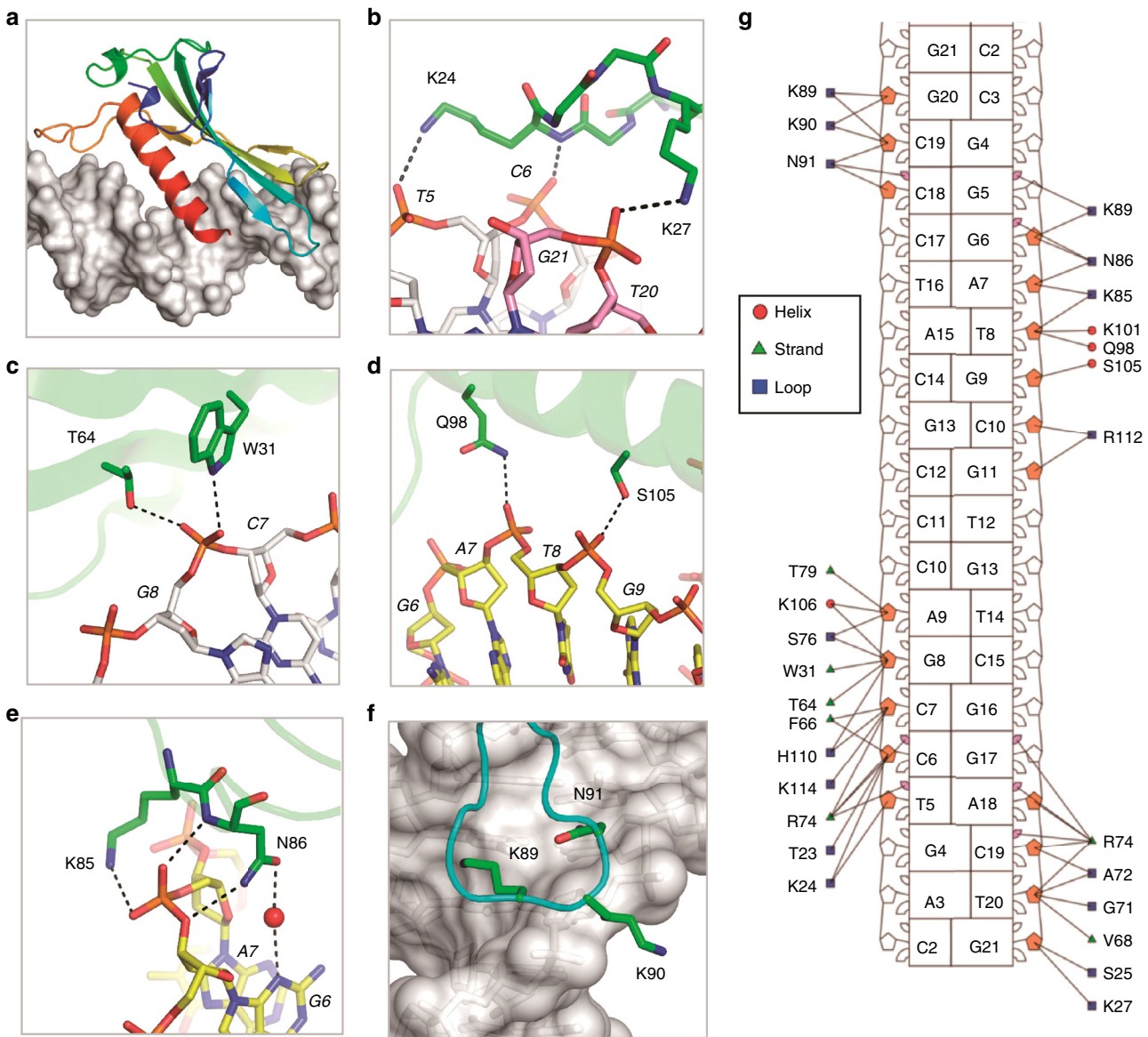

**Fig. 3** The interaction between DNA and *Asfv*LIG NTD. **a** Cartoon-and-surface view showing the relative orientation between *Asfv*LIG NTD and DNA. **b**–**f** Detailed interactions between *Asfv*LIG NTD and DNA residues. *Asfv*LIG NTD residues are shown as sticks in atomic colors (C, green; N, blue; O, red); the DNAs are also shown as sticks, the C-atoms of the template strand, upstream and downstream of the broken strand are colored in white, yellow, and pink, respectively. **g** Nucleotide-residue contact map showing individual nucleotide-residues interactions for the preferred binding site. Small and large markers on each nucleotide represent the major and minor groove contacts, respectively. Filled-in pink markers highlight which nucleotides are contacted by at least one residue in the minor groove

domain previously[23]. Replacing residues 85–92 of the DNA-interacting β7-α3 loop with two Gly residues (for *Asfv*LIG LD, Supplementary Fig. 6d) lowered the DNA-binding affinity by 2-fold. Deletion of the NTD domain (for *Asfv*LIG ΔN) completely abolished the DNA-binding ability of the protein (Supplementary Fig. 6d).

In addition to binding activity, we also investigated the DNA ligation activity of WT *Asfv*LIG and mutants using nick DNA-CG. As depicted in Fig. 4f and Supplementary Fig. 7a, the WT *Asfv*LIG can efficiently catalyze the ligation reaction, the reaction rate ($K_{obs}$) is $134.83 \pm 3.32 \times 10^{-6}$ min$^{-1}$; no ligation activity was observed for *Asfv*LIG ΔN (Supplementary Fig. 7b). *Asfv*LIG LD can support the ligation process (Supplementary Fig. 6e); consistent with its weak DNA-binding affinity, the reaction rate ($1.48 \pm 0.20 \times 10^{-6}$ min$^{-1}$) of *Asfv*LIG LD is significantly lower than that of the WT *Asfv*LIG. Together, these observations

indicate that NTD plays important role in DNA binding; the cooperative interactions between NTD and other domains (especially the AD domain) can enhance the DNA-binding affinity and are critical for the catalytic complex assembly and catalysis of *Asfv*LIG.

**Conformational comparison of the nick site base pairs.** Consistent with a previous study[13], our in vitro catalytic assays further confirmed that *Asfv*LIG is an error-prone DNA ligase (Fig. 1b and Supplementary Fig. 2). *Asfv*LIG can ligate various substrates with mismatched base pairs, as well as Watson–Crick paired DNA substrates (Supplementary Table 2). Our structures represent several different states of the reaction: the *Asfv*LIG:CG complex structure captured the Watson–Crick C:G pair prior to ligation (Supplementary Fig. 8a), whereas *Asfv*LIG:CT1 and

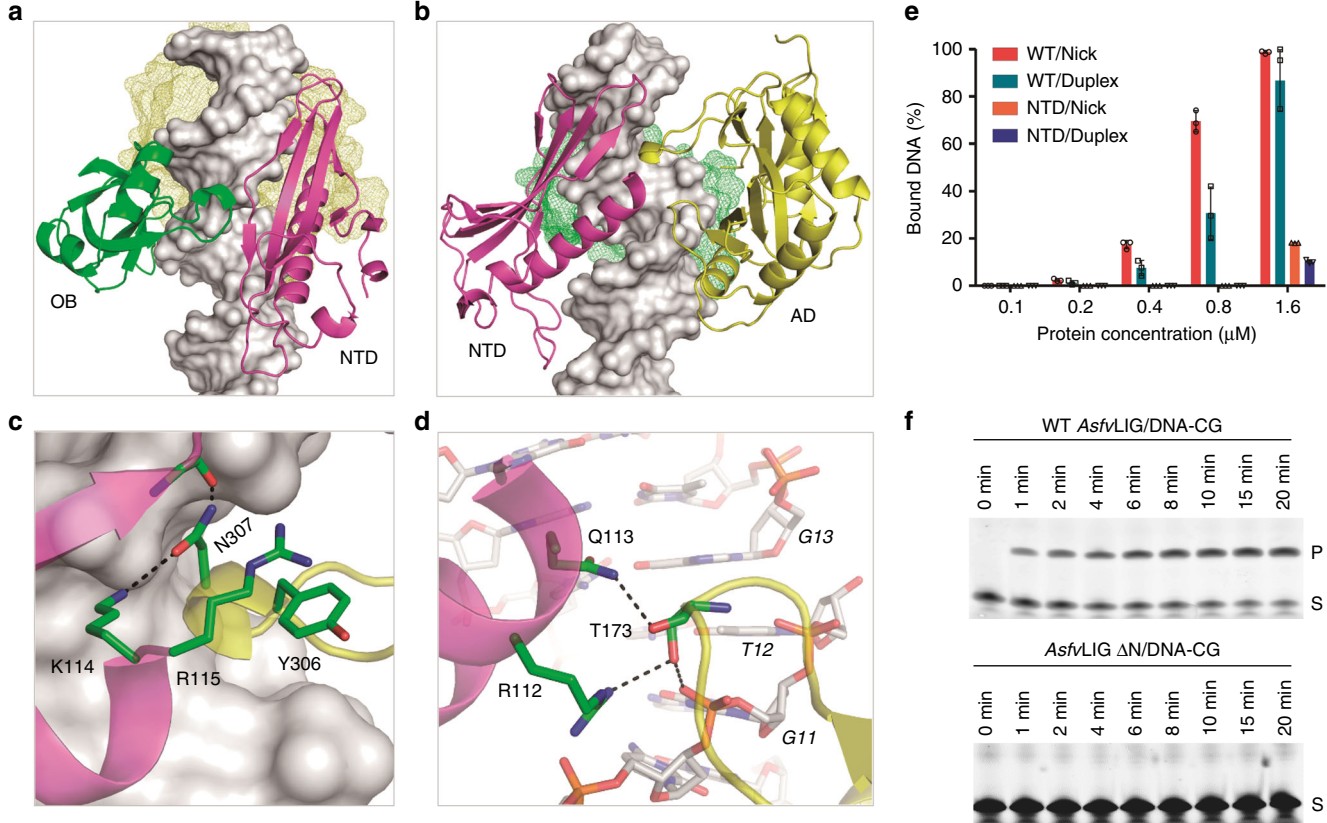

**Fig. 4** Impacts of *Asfv*LIG NTD on DNA binding and ligation. **a** Relative orientation between the NTD and OB domains in the catalytic form *Asfv*LIG-DNA complex. **b**–**d** Relative orientation and detailed interactions between the NTD and AD domains in the catalytic form structure. **e** Comparison of nick and duplex DNA-CG binding by WT *Asfv*LIG and *Asfv*LIG NTD. All data points from three independent experiments are shown with the median expressed as bars. The standard deviation (±SD) values are indicated by error bars. **f** In vitro DNA-CG ligation catalyzed by WT *Asfv*LIG and *Asfv*LIG with NTD deletion (for *Asfv*LIG ΔN). The substrate and product bands are labeled as S and P, respectively. Uncropped gels are shown in Supplementary Fig. 7

*Asfv*LIG:CT2 captured the mismatched C:T pair prior to and after the ligation process (Supplementary Figs. 8b, c), respectively.

DNA-CT is one of the mismatched substrates that can be ligated by *Asfv*LIG as efficiently as the Watson–Crick paired substrates. Analysis and comparison of the three catalytic structures revealed some potential basis underlying the C:T mismatch tolerance by *Asfv*LIG. In the *Asfv*LIG:CT1 structure (Fig. 5a and Supplementary Fig. 8d), the nucleobases of the template C11 and the nick site T12 form one H-bond (3.1 Å) between their N4 and O4 atoms, respectively. Further, via the water-mediated H-bond, the O2 atom of the C11 nucleobase also interacts with the main chain N atom of Gln403 of the OB domain. Though not identical, the overall shapes and orientations of the C:T pair in *Asfv*LIG:CT1 and the C:G pair in *Asfv*LIG:CG are similar (Fig. 5b and Supplementary Fig. 8e). Structural superposition (Fig. 5c) further revealed that presence of mismatched C11:T12 pair does not cause obvious conformational difference on 3′-OH, 5′-P, or phosphodiester backbone at the active site, compared with the Watson–Crick C11:G12 pair.

In the *Asfv*LIG:CT2 structure (Fig. 5d), the template C11 and the sealed T12 form one H-bond (2.9 Å) between their nucleobases. However, though the pairing and overall shape of C11:T12 pairs are similar in the *Asfv*LIG:CT1 and *Asfv*LIG:CT2 structures, superposition reveals some subtle conformational changes at the local regions. Compared to the *Asfv*LIG:CT1 structure, the C11-T12 pair is shifted about 0.5 Å toward the main chain N atom of Gln403 in the *Asfv*LIG:CT2 structure, which may exclude the C11-interacting water molecule

(Supplementary Fig. 8f). Meanwhile, the side chain of Gln403 rotates clock wise around the CG-CD bond for about 15° in the *Asfv*LIG:CT2 structure (Fig. 5e), leading to the formation of one H-bond (2.9 Å) between the NE2 atom of Gln403 and the O2 atom of the T12 nucleobase (Fig. 5d), thereby shifting the conformation to the ligated product.

**Leu402 and Gln403 impact the ligation activity of *Asfv*LIG.** Gln403 of *Asfv*LIG corresponds to Phe872 in *Hs*LIG1; interestingly, the residues prior to Gln403 and Phe872 are also different in *Asfv*LIG and *Hs*LIG1, which have Leu402 and Arg871, respectively. As depicted in Supplementary Fig. 1c, both Arg871 and Phe872 residues are highly conserved in *Hs*LIG1 and other DNA ligases[26,27,30]. Though Leu402 does not form strong interactions with the C11:T12 pairs in the *Asfv*LIG:CT1 and *Asfv*LIG:CT2 structures, its side chain resides next to the neighboring C12:G11 pair from the minor groove side (Fig. 5f). Among AD and OB domains, six residues (Leu211, Ala215, and Asn219 of the AD domain; Tyr363, Leu402, and Gln403 of the OB domain) form contacts with DNA in the minor groove (Supplementary Fig. 4a); Leu402 and Gln403 are the only two residues next to the nick of the DNA.

To investigate the potential roles of Leu402 and Gln403, we first constructed one *Asfv*LIG protein with the whole OB domain deleted (referred to as ΔOB hereafter). As depicted in Supplementary Fig. 9a, *Asfv*LIG ΔOB can bind both duplex and nick DNAs. The DNA-binding affinity of *Asfv*LIG ΔOB is stronger than that of *Asfv*LIG NTD but is much weaker than the

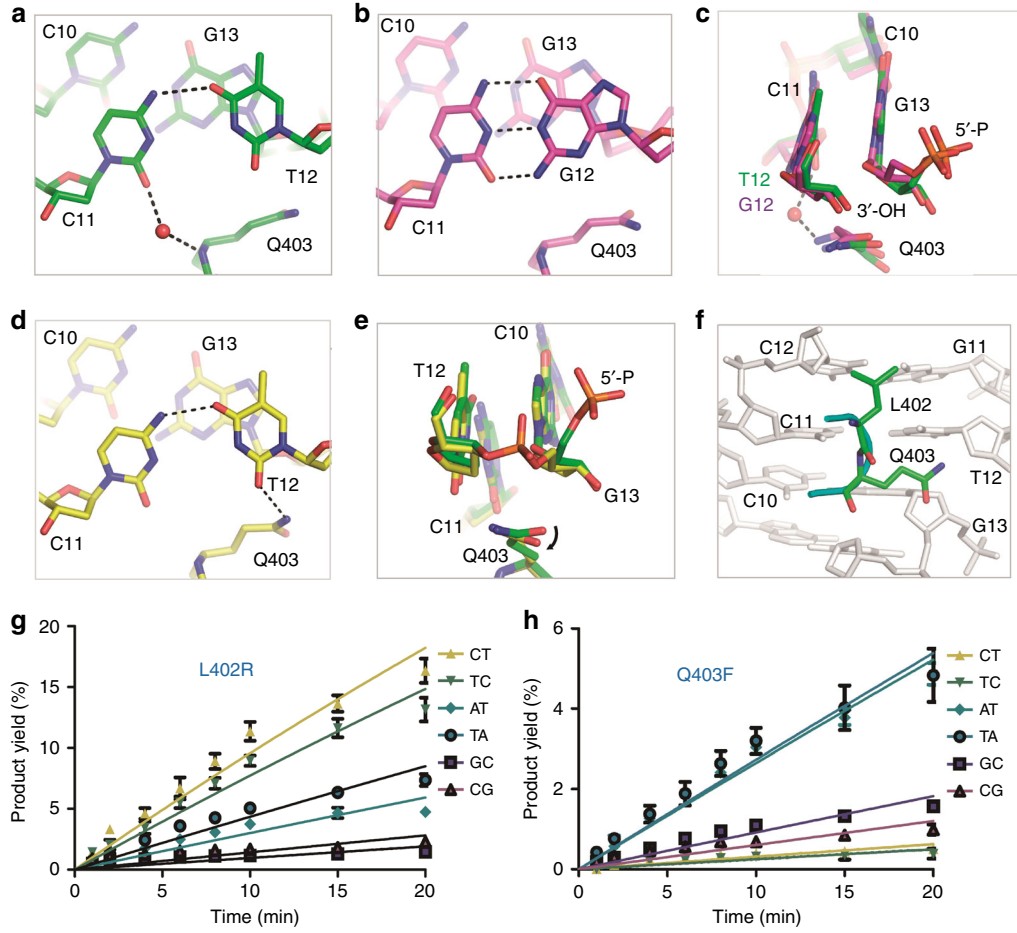

**Fig. 5** Structural basis for C:T and C:G base pair recognition by *Asfv*LIG. **a, b** Local conformations of the C:T and C:G pairs captured in the *Asfv*LIG:CT1 and *Asfv*LIG:CG structures, respectively. **c** Superposition of the nick site base pairs in *Asfv*LIG:CT1 and *Asfv*LIG:CG. **d** Local conformations of the C:T pair captured in the *Asfv*LIG:CT2 structure. **e** Superposition of the nick site base pairs in *Asfv*LIG:CT1 and *Asfv*LIG:CT2. **f** Detailed conformations of the active site residues Leu402 and Gln403, based on the *Asfv*LIG:CT1 structure. **g, h** In vitro DNA ligation catalyzed by L402R and Q403F mutants, respectively. The data represent the mean of three independent experiments, the standard deviation (±SD) values are indicated by error bars. In panels (**a–e**), the C-atoms of protein and DNA residues are colored in green, yellow, and magenta for the *Asfv*LIG:CT1, *Asfv*LIG:CT2, and *Asfv*LIG:CG structures, respectively

full-length WT *Asfv*LIG (Supplementary Fig. 6c). Unlike WT *Asfv*LIG, which mainly forms complex with DNA with a molar ratio of 1:1, ΔOB can form both 1:1 and 2:1 complex with DNA. However, as revealed by the in vitro ligation assay (Supplementary Fig. 9b), ΔOB could not support the ligation reaction, indicating that the observed ΔOB:DNA complexes are incompatible with the catalysis.

Besides *Asfv*LIG ΔOB, we also constructed two *Asfv*LIG mutants (L402R and Q403F) and carried out in vitro DNA ligation assays using DNA-CT and DNA-TC, the two most tolerable mismatched substrates of *Asfv*LIG. Compared to WT *Asfv*LIG (Fig. 1b), replacing Leu402 with Arg402 (for L402R mutant) lowered the DNA-CT and DNA-TC ligation rates of the protein by about 20-fold (Fig. 5g and Supplementary Fig. 10a), to $10.06 \pm 0.36 \times 10^{-6}$ min$^{-1}$ and $8.03 \pm 0.26 \times 10^{-6}$ min$^{-1}$, respectively; replacing Gln403 with Phe403 (for Q403F mutant) caused more than 600-fold reduction on the DNA-CT and DNA-TC ligation rates of the protein (Fig. 5h and Supplementary Fig. 10b), to only $0.31 \pm 0.02 \times 10^{-6}$ min$^{-1}$ and $0.25 \pm 0.03 \times 10^{-6}$ min$^{-1}$, respectively.

As noted previously, mismatched C:T pair and Watson–Crick C:G pair have similar conformations in the *Asfv*LIG:CT1 and *Asfv*LIG:CG structures (Supplementary Fig. 8e). To test whether Leu402 and Gln403 impact the ligation efficiency of

Watson–Crick paired substrates, we performed in vitro ligation assays using DNA-CG, DNA-GC, DNA-AT, and DNA-TA (Fig. 5g, h). Compared to the WT *Asfv*LIG, the L402R and Q403F mutants had much lower ligation rates, varied from ~40–75-fold for DNA-AT and DNA-TA substrates to ~100–200-fold for DNA-GC and DNA-CG substrates. In general, the impact of the Q403F mutation was greater than that of L402R mutation toward these DNA substrates.

**Comparison with homologous proteins**. The AD and OB domains are conserved in *Asfv*LIG and all *Hs*LIG1-4 proteins. As depicted in Supplementary Fig. 1b, *Asfv*LIG AD shares 38%, 36%, and 29% sequence similarity with the AD domains of *Hs*LIG1, *Hs*LIG3, and *Hs*LIG4, respectively. Compared to the AD domains, the sequence similarity between the OB domains (Supplementary Fig. 1c) are lower between *Asfv*LIG and *Hs*LIG1, *Hs*LIG3, and *Hs*LIG4, which are 18%, 17%, and 20%, respectively. In addition to *Hs*LIG1:DNA complex, one close catalytic form *Hs*LIG3:DNA complex (PDB_ID: 3L2P)[24] and several open form *Hs*LIG4 structures[31] have been reported. The overall structures of *Hs*LIG1:DNA and *Hs*LIG3:DNA complexes are very similar. Superposition between *Asfv*LIG1:CT1 and *Hs*LIG1:DNA (Supplementary Fig. 5c) or *Hs*LIG3:DNA (Supplementary Fig. 11a) resulted in rmsd values of 1.9 Å in their AD domains, based on

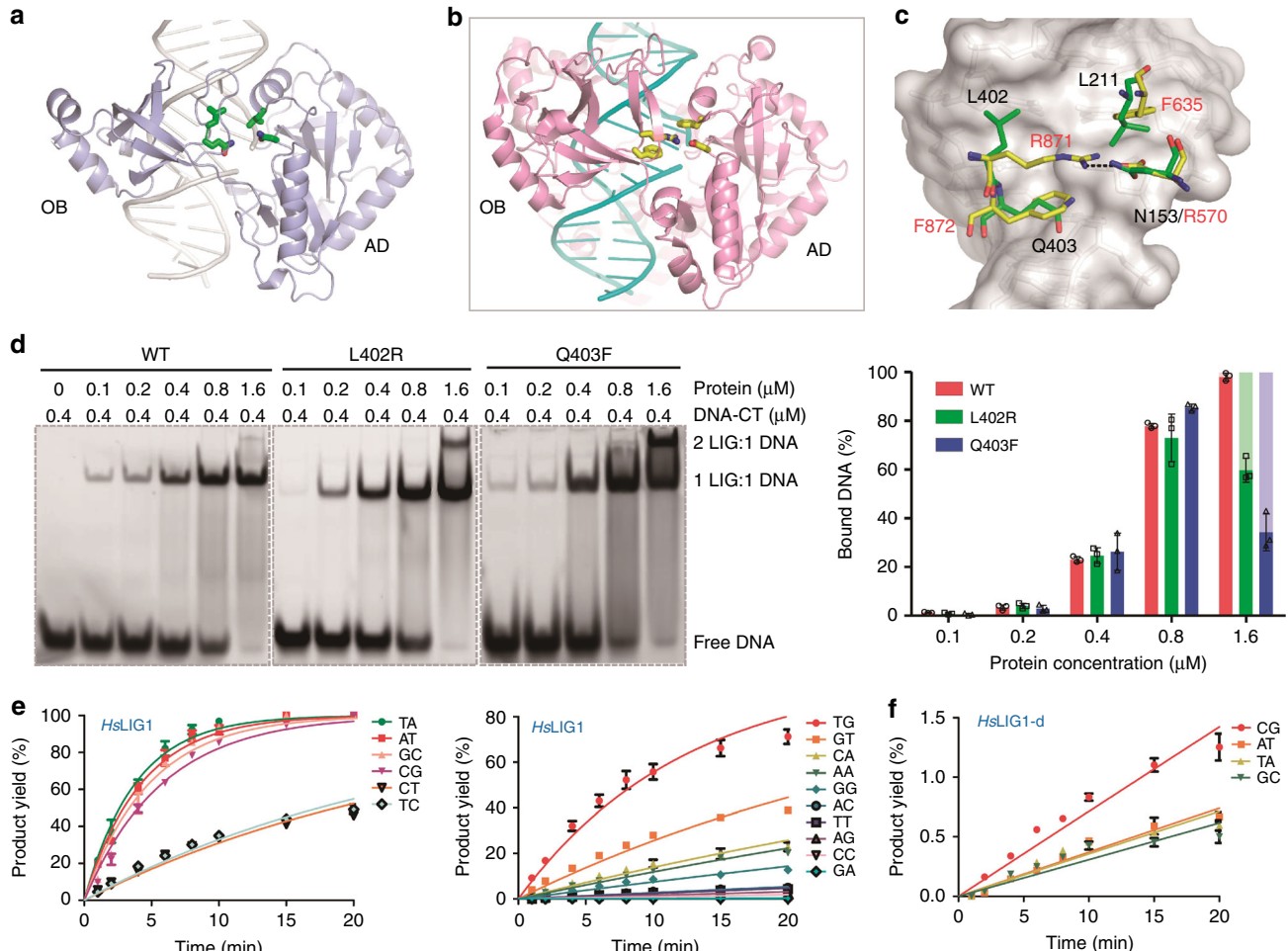

**Fig. 6** Unique ligase residues in proximity of the nick in the substrate DNA of *Asfv*LIG. **a**, **b** Cartoon representation showing the DNA-binding surface formed by the AD and OB domains of *Asfv*LIG and *Hs*LIG1, respectively. **c** Superposition of the ligase residues in proximity of the nick in the substrate DNA. In (**a–c**), all the ligase residues are shown as sticks. Identical colors are utilized for the N-atoms (blue) and O-atoms (red) in both structures, but the C-atoms are colored in yellow and green in *Asfv*LIG and *Hs*LIG1, respectively. **d** Comparison of the in vitro DNA-CT binding by WT *Asfv*LIG, and L402R and Q403F mutants. In the right panel, WT *Asfv*LIG, and L402R and Q403F mutants are colored in red, green, and blue, respectively. The complexes with protein:DNA molecular ratio of 2:1 are colored in light-green or light-blue. All data points from three independent experiments are shown with the median expressed as bars. The standard deviation (±SD) values are indicated by error bars. Uncropped gels are shown in Supplementary Fig. 13. **e** In vitro DNA ligation catalyzed by WT *Hs*LIG1. **f** In vitro DNA ligation catalyzed by *Hs*LIG1-d mutant. *Hs*LIG1-d and *Hs*LIG1-q stand for R871L/F872Q double mutant and D570N/F635L/R871L/F872Q quadruple mutant, respectively. The data represent the mean of three independent experiments, and the standard deviation (±SD) values are indicated by error bars

the superposition of 166 and 173 pairs of Cα atoms. Possibly due to the lower sequence similarity, *Asfv*LIG1 OB domain exhibits more obvious conformational differences with *Hs*LIG1 and *Hs*LIG3, especially for the α-helix located at the edge. Based on the superposition of 82 and 74 pairs of Cα atoms, the rmsd values between the OB domains of *Asfv*LIG1 and *Hs*LIG1 and *Hs*LIG3 are 2.9 ÅÅ and 2.8 Å, respectively.

All the *Hs*LIG4 structures adopt an open conformation (Supplementary Fig. 11b). The relative orientations between the AD and OB domains of *Hs*LIG4 structures are different from our open form *Asfv*LIG:DNA complex (Fig. 1c and Supplementary Fig. 5a). However, structural comparison showed that the overall conformations of the AD domains of *Hs*LIG4 and *Asfv*LIG are similar (Supplementary Fig. 11c); the rmsd value between them is 1.8 Å, based on the superposition of 159 pairs of Cα atoms. Compared to *Hs*LIG1 and *Hs*LIG3, the OB domain of *Asfv*LIG showed more significant conformational difference to *Hs*LIG4 (Supplementary Fig. 11d). Superposition of the central β-sheet

core resulted in a rmsd value great than 3.0 Å; all α-helice took significant different locations in the two structures.

In addition to Leu402 and Gln403 (or Arg871 and Phe872) of the OB domain, the above comparison revealed that sequences and conformations of several AD domain residues surrounding the DNA nick sites are different in *Asfv*LIG and *Hs*LIG1 (Fig. 6a–c). Instead of the neighboring base pair, Arg871 of *Hs*LIG1 is located directly under the nick site base pair and forms one salt bridge with Asp570. The Arg871-Asp570 pair is flanked by Phe872 on one side and by another Phe residue (Phe536) on the opposite side. Besides DNA distortion, previous studies suggested that the Asp570-Arg871 salt bridge and packing interactions between the Phe residues (Phe872 and Phe635) and the ribose sugars of nucleotides on the 5′ and 3′ ends of the nick all contribute to the fidelity and efficiency of *Hs*LIG1[23]. Both Asp570 and Phe635 of the *Hs*LIG1 AD domain are highly conserved in many DNA ligases, whereas they are replaced by Asn153 and Leu211 in *Asfv*LIG, respectively (Supplementary Fig. 1b).

The conformations of Asn153 and Asp570 are similar in the two structures, whereas the conformations of Leu211, Leu402, and Gln403 of *Asfv*LIG are different from the corresponding residues of *Hs*LIG1. To further investigate the functions of these unique residues, we carried out in vitro ligation assays using L402R/Q403F double mutant, N153D/L402R/Q403F triple mutant, and N153D/L211F/L402R/Q403F quadruple mutant (Supplementary Fig. 12 and Supplementary Table 3). Interestingly, none of these *Asfv*LIG mutants showed enhanced catalytic efficiency toward any substrates, including Watson–Crick paired and mismatched DNAs. In contrast, the activities of the three mutants were much lower than the WT *Asfv*LIG; and, for most of the substrates, their ligation activities were even lower than the L402R and Q403F mutants.

Together with the mutants with a single mutation, the ligation assay results of the double, triple, and quadruple mutants indicated that the four nick site residues play important roles in the catalytic efficiency of *Asfv*LIG. Compared to the Asp570-Arg871 salt bridge and the aromatic rings of Phe635 and Phe872 observed in the *Hs*LIG1 structure, the local conformation formed by Asn153, Leu211, Leu402, and Gln403 were more flexible in the *Asfv*LIG structure; instead of discrimination between Watson–Crick paired and mismatched base pairs, our ligation results indicate that these residues may play a more important role in the assembly and stabilization of the catalytic complex. To further support this hypothesis, we carried out in vitro DNA-binding assays (Fig. 6d and Supplementary Fig. 13). Compared with WT *Asfv*LIG, replacing either Leu402 with Arg402 or Gln403 with Phe403 at low concentrations (ranging from 0.1 to 0.8 μM) has no clear impacts on DNA binding. At high concentration (1.6 μM), the overall DNA-CT binding abilities of L402R and Q403F were comparable to that of WT *Asfv*LIG; however, in contrast to WT *Asfv*LIG, a significant number of slower moving bands were observed on the native gel in the presence of mutants, especially Q403F. We speculate that the slower moving band may correspond to a complex with a protein: DNA molecular ratio of 2:1; similar assembly has been observed in the non-catalytic form *Asfv*LIG:DNA structure (Supplementary Fig. 5a).

**Switching of nick site residues between *Asfv*LIG and *Hs*LIG1.** The four *Hs*LIG1 residues (Asp570, Phe635, Arg871, and Phe872) in the proximity of the nick in the DNA substrate are highly conserved in many DNA ligases. However, our in vitro ligation assays clearly indicated that these residues can not replace the corresponding residues of *Asfv*LIG in catalysis. To further investigate the functional role of these nick site residues, we also constructed and purified the WT *Hs*LIG1, one R871L/F872Q double mutant (*Hs*LIG1-d), and one D570N/F635L/R871L/F872Q quadruple mutant (*Hs*LIG1-q) of *Hs*LIG1. During in vitro ligation assays, we found that the overall ligation activity of WT *Hs*LIG1 is higher than that of *Asfv*LIG. The ligation rates of Watson–Crick paired DNAs are comparable in the presence of 0.01 μM *Hs*LIG1 (Fig. 6e, Supplementary Fig. 14, and Supplementary Table 4) or 0.05 μM *Asfv*LIG (Fig. 1b and Supplementary Fig. 2). Among the non-Watson–Crick paired substrates, *Hs*LIG1 can catalyze the ligation of DNA-TG most efficiently, but the DNA-TG ligation rate ($K_{obs}$) is 2–3 fold lower than those of Watson–Crick paired DNAs. A similar phenomenon has been observed previous[32], maybe due to the T:G wobble-pair formation in the nick site. Unlike WT *Hs*LIG1, *Hs*LIG1-d could not catalyze the ligation of any non-Watson–Crick paired DNA substrate (Supplementary Fig. 15). Though *Hs*LIG1-d can still catalyze the ligation of Watson–Crick paired DNA substrates (Fig. 6f, Supplementary Fig. 15, and Supplementary Table 4), the

reaction rate is much lower than those of WT *Hs*LIG1. *Hs*LIG1-q could not catalyze the ligation of any Watson–Crick or non-Watson–Crick paired DNA substrates (Supplementary Fig. 16 and Supplementary Table 4). Though we are not sure about their exact role in the fidelity of *Hs*LIG1, our in vitro ligation assay results clearly indicated that these nick site residues play important role in the catalytic efficiency of *Hs*LIG1.

**Catalytic mechanism and in vitro nick DNA binding.** Despite their differences in sequence and size, all the characterized ATP-dependent DNA ligases catalyze phosphodiester bond formation between adjacent 3′-OH and 5′-phosphate in DNA duplex through a similar three-step mechanism (Fig. 7a). In the first step, the $NH_2$ group of the catalytic Lys residue attacks the α-phosphate of ATP, forming an enzyme-AMP intermediate and inorganic pyrophosphate (PPi). In the second step, nick DNA binds to the enzyme-AMP intermediate and the 5′-P of the downstream DNA attacks Lys-AMP to form a pyrophosphate linked AppDNA intermediate. In the third step, the 3′-OH of the upstream DNA attacks the pyrophosphate group of AppDNA, covalently joining the DNA strands and liberating AMP. The ATP was captured in the active site of the non-catalytic form *Asfv*LIG:DNA structure (Fig. 1c, d). The catalytic Lys residue (Lys151) and ATP-interacting residues of *Asfv*LIG are highly conserved in the homologous proteins (Supplementary Fig. 1). Though the domain arrangement of the non-catalytic form *Asfv*LIG:DNA complex is very different from that of *Hs*Lig1 complexed with adenylated DNA[23], the interactions between ATP and the interacting residues are very similar in the two structures (Fig. 7b), indicating that *Asfv*LIG follows a conserved mechanism in ATP binding and catalysis.

Conceptually, differentiation between matched and mismatched base pairs first takes place in step 2 of the reaction. To verify whether *Asfv*LIG discriminates between match and mismatch during step 2, we carried out in vitro DNA-binding assay (Fig. 7c and Supplementary Fig. 17). Similar to the four Watson–Crick paired DNAs, DNAs with mismatched base pairs at the 3'-end of the nick can all be efficiently bound by *Asfv*LIG; the binding affinities between *Asfv*LIG and all DNAs are very similar. Besides *Asfv*LIG, we also carried out in vitro DNA-binding assay using *Hs*Lig1 (Fig. 7d and Supplementary Fig. 18). Compared to *Asfv*LIG, *Hs*Lig1 has similar binding affinity to mismatched DNAs with pyrimidines (C or T) on the template strand; however, when the protein concentrations are within the range of 0.2–0.8 μM, the binding affinities between *Hs*Lig1 and mismatched DNAs with purines (A or G) on the template strands are significantly weaker than those of *Asfv*LIG. Very surprising, compared to the mismatched DNAs, the binding affinities between *Hs*Lig1 and the four matched DNAs are much weaker.

**Discussion**
ASFV is contagious and can cause lethal diseases in both domestic pigs and wild boars. *Asfv*LIG catalyzes the DNA ligation reaction during the base excision repair (BER) process of ASFV genome. However, unlike the homologous proteins (such as *Hs*LIG1 and *Sus*LIG1), the fidelity of *Asfv*LIG is very low. By solving four *Asfv*LIG:DNA complex structures, including the non-catalytic open form (Fig. 1c) and the catalytic closed forms (Fig. 2a), we showed that the NTD, AD, and OB domains of *Asfv*LIG undergo a large conformational change during the catalytic assembly. A bound ATP molecule is present in the non-catalytic structure (Fig. 1c and Supplementary Fig. 5a), demonstrating that *Asfv*LIG has ATP binding and catalytic mechanism commonly shared by many other ligases (Supplementary Fig. 5b)[33–36].

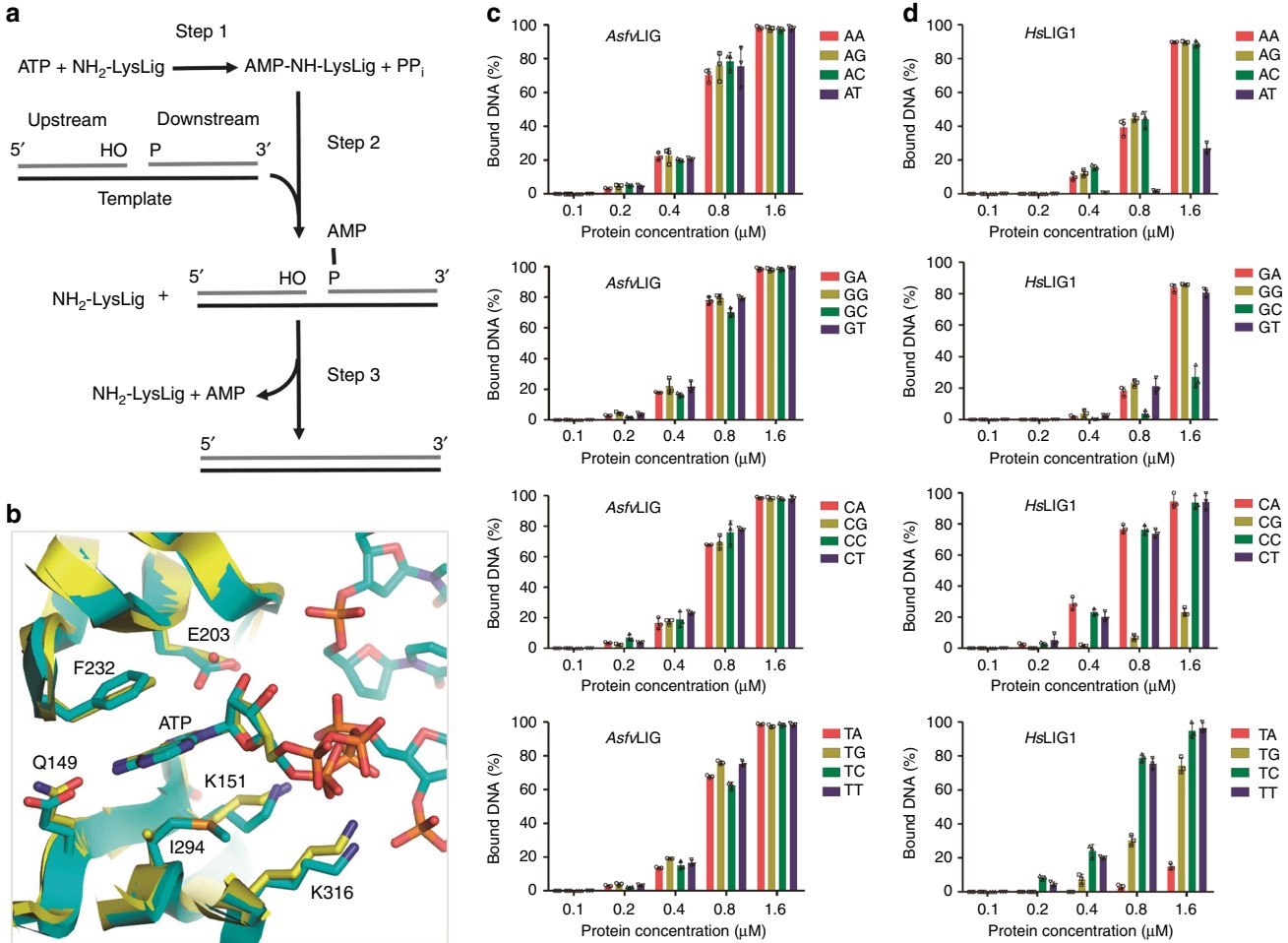

**Fig. 7** Comparison of ATP and DNA binding by *Asfv*LIG and *Hs*LIG1. **a** The conserved catalytic mechanism of ATP-dependent DNA ligation. **b** Superposition showing the similar ATP binding observed in *Asfv*LIG and *Hs*LIG1 structures. *Asfv*LIG of the non-catalytic *Asfv*LIG:DNA complex is shown as cartoon in yellow, and ATP and ATP-interacting residues are shown as sticks in atomic colors (C, yellow; N, blue; O, red; P, orange). *Hs*LIG1 in the complex with adenylated DNA (PDB_ID: 1 × 9 N) is shown as cyan cartoon. DNA, AMP, and AMP-interacting residues of *Hs*LIG1 are shown as sticks in atomic colors (C, cyan; N, blue; O, red; P, orange). **c** In vitro DNA binding by *Asfv*LIG1. **d** In vitro DNA binding by *Hs*LIG1. All data points from three independent experiments are shown with the median expressed as bars. The standard deviation (±SD) values are indicated by error bars

In the catalytic form *Asfv*LIG:DNA complex structures, *Asfv*-LIG NTD domain mimics the DBD domains of the homologous ligases; however, *Asfv*LIG NTD has a complete novel DNA-binding fold. Deletion of NTD (for *Asfv*LIG ΔN) completely abolished the DNA binding (Supplementary Fig. 6d) and ligation (Fig. 4f) activities of *Asfv*LIG. Clearly, the extensive interactions between *Asfv*LIG NTD and nick DNAs contribute to the strong DNA-binding affinity of *Asfv*LIG, which are equal for matched and mismatched DNAs (Fig. 7c). *Hs*Lig1 is a high fidelity DNA ligase (Fig. 6e). Based on the *Hs*Lig1:DNA complex structure (PDB_ID: 1 × 9 N), it has been previously proposed that *Hs*Lig1 can cause some local distortion on the duplex; this distortion is important for the alignments of the 3′-OH and the adenylated 5′-P for nick sealing[23]. The overall matched DNA ligation activity of *Hs*Lig1 is much higher than that of *Asfv*LIG. However, very surprisingly, *Hs*Lig1 has much weaker binding affinities towards the matched nick DNAs, compared to the mismatched nick DNAs (Fig. 7d). These observations suggested that *Hs*Lig1 may be able to sense the structural perturbation at the position of the mismatch and stay bound to the DNA, leading to ligation inhibition.

Interestingly, though the overall folds of the AD and OB domains are conserved in *Asfv*LIG and the homologous proteins,

their sequence identities are very low (Supplementary Figs. 1b, c). At the active site, *Asfv*LIG contains four unique residues, including Asn153 and Leu211 of the AD domain and Leu402 and Gln403 of the OB domain. Compared to *Asfv*LIG, the local conformation formed by the corresponding nick site residues (including Asp570, Phe635, Arg871, and Phe872) of *Hs*Lig1 is much more rigid. The nick site residues could not be switched between *Asfv*LIG and *Hs*Lig1, suggested that these residues might work in a cooperative manner with their corresponding NTD or DBD domain only. Though they are not involved in the direct catalytic process, our in vitro DNA binding and ligation assay results (Figs. 5g, h, 6d, 7b, and Supplementary Figs. 10, 12, 14-16) suggested that these nick site residues are important for nick recognition and thus catalytic efficiency of *Asfv*LIG and *Hs*Lig1.

Consistent with a previous study[37], the obvious different DNA-binding behavior, unique fold of the NTD domain, and unique residues at the nick site all confirm that *Asfv*LIG is one atypical DNA ligase. Before any *Asfv*LIG structure was reported, Showalter and co-workers already demonstrated that *Asfv*LIG has very low adenylation activity towards DNAs with 3′-dideoxy- or 3′-amino-terminated nicks, compared to regular nick DNAs;[13] these observations indicated that 3′-OH of the nick is a critical component of the active site architecture during 5′-P adenylation.

The interaction between the NTD domain and the four nick site residues will help the reorientation of DNA 3′-OH and facilitate the ligation reaction. Compared to the other three nick site residues, Gln403 appears to be more important for the catalytic efficiency of *Asfv*LIG, maybe due to its H-bond interaction with the nick site base pairs (Fig. 5b). At present, the exact molecular basis underlying the low fidelity of *Asfv*LIG still remains elusive. More biochemical and structural data, especially the structures of *Asfv*LIG complexed with adenylated DNA, are required to better understand the tolerance of mismatched DNAs during the catalytic process.

Recent discovery of the poly (ADP-ribose) polymerase inhibitor (which can selectively target the DNA repair defect in hereditary breast cancer) has stimulated the inhibitor development of other repair enzymes[38]. As the enzyme required during almost all the repair events, ligase has been most extensively targeted[39–41]. Obvious repair inhibitory effects have been observed for many compounds targeting the DBD domains of human ligases (including *Hs*LIG1-4) in vitro and/or in vivo[42]. In combination with their subtoxicities, some compounds can preferentially sensitize human cancer cell lines to the cytotoxic effects of other DNA agents. However, due to the conservation of DBD domains, many inhibitors can simultaneously target various human ligases, which limited their therapeutic potential.

ASFV is replicated and assembled in swine macrophage cells, which are oxidative and cause continuous damage to the virus genome. Though it is error-prone and may contribute to quick genotype formation, *Asfv*LIG is the sole ligase involved in the repair pathway. Therefore, inhibiting its activity will certainly disrupt the repair process and impair the genome stability of ASFV. *Asfv*LIG NTD has a mixed α/β fold in nature and represents a complete novel DNA-binding motif; though it may be not important for the fidelity, *Asfv*LIG NTD plays critical role in substrate binding and catalytic complex assembly of *Asfv*LIG. Although the structures of pig DNA ligases, including *Sus*LIG1-4, have not been characterized, they should be very similar to those of human *Hs*LIG1-4 evidenced by the high sequence similarities (all around 90%). Since *Asfv*LIG NTD has no sequence or folding similarity with any DNA ligases in pigs, it can serve as an excellent target for the development of small molecule inhibitor, and the *Asfv*LIG NTD targeting inhibitors will not interfere with the normal DNA joining functions of the pig ligases. Besides NTD, the nick site residues (Asn153, Leu211, Leu402, and Gln403) of *Asfv*LIG are very unique. As revealed by our mutagenesis and in vitro catalytic studies, these nick site residues cannot be switched between *Asfv*LIG and the homologous proteins. Small molecules targeting the nick site residues of *Asfv*LIG and other unique features of ASFV repair enzymes, such as the 5′-P binding pocket in the finger domain of *Asfv*PolX, can all impair ASFV repair process and help provide tools to combat this deadly virus.

## Methods

**Plasmid construction.** The gene containing the codon-optimized cDNA sequence of full-length WT *Asfv*LIG (Supplementary Table 5) was purchased from Shanghai Generay Biotech Co., Ltd, China. The gene was cleaved with BamHI and XhoI and visualized using an agarose gel. The target fragment was recovered and recombined into the pET28-Sumo vector. The recombinant vector (coding for His-Sumo-*Asfv*LIG) was then transformed into *Escherichia coli* BL21 DE3 competent cells for protein expression. The recombinant His-Sumo-*Asfv*LIG coding vector was utilized as the template during the plasmid constructions of all *Asfv*LIG NTD, *Asfv*LIG ΔN, *Asfv*LIG LD (in which the residues 85–92 of WT *Asfv*LIG were replaced by two Gly residues), L402R, and Q403F mutants, via overlap polymerase chain reactions (PCR) or site direct mutagenesis according to the manufacturer's protocols. The resulting His-Sumo-*Asfv*LIG L402R plasmid DNA was then used as the template for constructing double (L402R/Q403F), triple (L402R/Q403F/N153D), and quadruple (L402R/Q403F/N153D/L211F) mutants. Detailed sequences of the primers used in the constructions are listed in Supplementary Table 5.

The region coding for residues 262 to 919 of *Hs*LIG1 was amplified by PCR using human genomic cDNA as template and sub-cloned into the BamHI/XhoI restriction sites of pET28a- Sumo vector. The resulting recombinant vector was transformed into the *Escherichia coli* BL21 DE3 competent cells and the plasmid DNA was extracted and used as template for the R871L/F872Q double mutant construction with site direct mutagenesis kit. The R871L/F872Q plasmid DNA was then used in the preparation of the *Hs*LIG1 D570N/F635L/R871L/F872Q quadruple mutant. Detailed sequences of the primers used in WT and mutant *Hs*LIG1 constructions are listed in Supplementary Table 6. Sequences of all WT and mutant of His-Sumo-*Asfv*LIG and His-Sumo-*Hs*LIG1 plasmids were confirmed by DNA sequencing. All recombinant strains were preserved using 30% glycerol and stored in a −80 °C freezer prior to use.

**Protein expression and purification.** All His-Sumo-*Asfv*LIG and His-Sumo-*Hs*LIG1 proteins were expressed using the same procedures. Briefly, the frozen recombinant strains were revived in Lysogeny broth (LB) medium supplemented with 50 μg/mL kanamycin at 37 °C overnight. Every 20 mL revived bacterium suspension was inoculated into 1 L LB medium supplemented with kanamycin (50 μg/mL) and cultured at 37 °C with continuous shaking. Protein expression was induced at $OD_{600} ≈ 0.6$ by adding of isopropyl β-D-1-thiogalacto-pyranoside (IPTG) at a final concentration of 0.1 mM. The induced cultures were then grown at 18 °C for an additional 18 h. The cells were harvested by centrifugation.

For overproduction of the Se-Met substituted *Asfv*LIG, the revived recombinant strains from 20 mL overnight cultures were inoculated into 1 L LB medium supplemented with 50 μg/mL kanamycin and grown at 37 °C. When $OD_{600}$ reached 0.4, the cells were harvested by centrifugation and resuspended in 100 mL M9 medium ($47.7$ mM $Na_2HPO_4$, 22 mM $KH_2PO_4$, 8.6 mM NaCl, and 28.2 mM $NH_4Cl$). The resuspended cells were centrifuged and transferred into 900 mL fresh M9 medium supplemented with 50 μg/mL kanamycin and 30 mg/L Se-Met (J&K). After growing at 37 °C for 1 h, the temperature was lowered to 18 °C and the protein expression was induced by addition of IPTG at a final concentration of 0.1 mM. The induced cultures were then grown at 18 °C for an additional 18 h and the cells were harvested by centrifugation.

All *Asfv*LIG proteins were purified using the same procedures. The cell pellets were resuspended in Buffer A (20 mM Tris pH 8.0, 500 mM NaCl, 25 mM imidazole pH 8.0) and lysed under high pressure via a JN-02C cell crusher. The homogenate was clarified by centrifugation and the supernatant was loaded onto a HisTrap™ HP column equilibrated with Buffer A. The fusion protein was eluted from the column using Buffer B (20 mM Tris pH 8.0, 500 mM NaCl, 500 mM imidazole pH 8.0) with a gradient. The fractions containing the desired fusion proteins were pooled and dialyzed against Buffer S (20 mM Tris pH 8.0, 500 mM NaCl, 25 mM imidazole pH 8.0) at 4 °C for 3 h; Ulp1 protease was also added to the sample during the dialysis process. The sample was again loaded onto the HisTrap™ HP column, the target protein was collected, and mixed with Tris buffer (20 mM, pH 8.0). The diluted sample was applied to a HiTrap SP HP column equilibrated with S binding buffer (20 mM Tris pH 8.0, 100 mM NaCl) and eluted using S elution buffer (20 mM Tris pH 8.0, 1 M NaCl) with a continuous gradient. The target protein was concentrated and loaded onto a Hi 16/60 Superdex G200 column equilibrated with gel filtration buffer (20 mM Tris pH 8.0, 100 mM NaCl). All *Hs*LIG1 proteins were purified using the similar procedures without the HiTrap SP HP column purification step. A volume of 1 mM DTT was present in all buffers, to prevent the oxidation of the proteins. The purity of the proteins was analyzed using an SDS-PAGE gel and the samples were stored at −80 °C until use.

**In vitro DNA binding and catalysis assays.** All DNAs used (Supplementary Table 1) in the binding and catalytic assays were purchased from Shanghai Generay Biotech Co., Ltd. The sixteen DNA substrates containing either Watson–Crick paired or mismatched base pairs were assembled by mixing the continuous template strand with upstream and downstream oligos in a molar ratio of 1:1:1 in gel filtration buffer. For catalysis assays, a 10-μL reaction system (composed of 6 μL gel filtration buffer, 1 μL 100 mM $MgCl_2$, 1 μL 10 mM ATP, 1 μL 8 μM DNA, and 1 μL protein) was established. The final *Asfv*LIG and *Hs*LIG1 protein concentrations are 0.05 μM and 0.01 μM, respectively. The reactions were carried out at 37 °C and quenched by the addition of 10 μL termination buffer (90% formamide, 20 mM EDTA, 0.05% bromophenol blue, and 0.05% xylene blue) at various time points. A total of 8 μL samples were loaded onto pre-warmed 18% urea sequencing gels and run at 18–20 W and 40–45 °C for 60 min. The gel was visualized using Typhoon FLA 9000, and intensities of the substrate and product bands were quantified by ImageQuantTL. Data were then fitted to the exponential $Y = Y_{max}[1-e^{(-K_{obs}t)}]$ using non-linear regression in GraphPad Prism 5. The observed rate constant ($K_{obs}$) and maximum ligation yield ($Y_{max}$) were determined from the regression curve.

Substrates DNA-CT or DNA-CG were diluted to 4 μM with gel filtration buffer and used in electrophoretic mobility shift assays (EMSA). A 10-μL reaction system, composed of 7 μL gel filtration buffer, 1 μL DNA (4 μM), and 2 μL proteins (either WT or mutants at a concentration of 8 μM, 4 μM, 2 μM, 1 μM, or 0.5 μM) was established. The samples were incubated at 0 °C for 90 mins and mixed with 10 μL loading buffer (20% glycerol, 0.05% bromophenol blue, and 0.05% xylene blue). A total of 8 μL samples were loaded onto pre-cooled 10% native gels and run at 120 V for 60 min. The gel was imaged using Typhoon FLA 9000. The intensities of the

bands were quantified by ImageQuantTL and the data were compared using GraphPad Prism.

**Crystallization and data collection**. All DNAs used in the structural studies were dissolved in ddH$_2$O; detailed sequences of the DNAs are listed in Supplementary Table 7. The crystallization samples were prepared by mixing *Asfv*LIG, DNA, and MgCl$_2$ and/or ATP (if present) at room temperature. The initial crystallization conditions for all crystals were identified at 18 °C using the crystallization robot system and commercial crystallization kits. During the initial screening, the sitting-drop vapor diffusion method was used, whereas all the crystal optimization procedures were performed using the hanging-drop vapor diffusion method. The compositions of the final crystallization conditions are also listed in Supplementary Table 7.

All crystals were cryoprotected using their mother liquor supplemented with 25% glycerol and snap-frozen in liquid nitrogen. X-ray diffraction data were collected on beamline BL17U and BL19U at the Shanghai Synchrotron Radiation Facility (SSRF). One crystal was used for each structure; data processing was carried out using the HKL2000 or HKL3000 programs[43]. The data collection and processing statistics are summarized in Supplementary Table 8.

**Structure determination and refinement**. The Se-*Asfv*LIG:DNA structure was solved using the single-wavelength anomalous diffraction (SAD) method[44] with the AutoSol program[45] embedded in the Phenix suite[46]. The Figure of Merit (FOM) value was 0.36. The initial model, which covers ~70% of protein residues in the asymmetric unit, was built using the Autobuild program and was refined against the diffraction data using the Refmac5 program[47] of CCP4i. During refinement, 5% of randomly selected data was set aside for free R-factor cross validation calculations. The 2F$_o$-F$_c$ and F$_o$-F$_c$ electron density maps were regularly calculated and used as guides for building the missing amino acids, DNA, and solvent molecules using COOT[48]. All *Asfv*LIG:CG, *Asfv*LIG:CT1, and *Asfv*LIG:CT2 structures were solved using the MR method with the Phaser program of the CCP4i suite using the individual domains of Se-*Asfv*LIG:DNA structure as the search model. The final refinement of all structures was done using the phenix.refine program[49] of Phenix. The structural refinement statistics are summarized in Supplementary Table 8.

**Reporting summary**. Further information on experimental design is available in the Nature Research Reporting Summary linked to this article.

## Data availability
Structural factors and coordinates have been deposited in the Protein Data Bank under accession codes 6IMJ, 6IMK, 6IML, and 6IMN for Se-*Asfv*LIG:DNA, *Asfv*LIG:CG, *Asfv*LIG:CT1, and *Asfv*LIG:CT2, respectively. All other data are available from the corresponding author upon reasonable request.

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

## Acknowledgements
We thank the BL17U and BL19U beamline staff at the Shanghai Synchrotron Radiation Facility (SSRF) for help during data collection. This work was supported by the National Natural Science Foundation of China (31870721, 31470724, 31670878, 21572146, 21761132029) and the Key Research and Development Project of China (2016YFA0500600), and Key R&D Program of Sichuan (2018NZ0151).

## Author contributions
Y.C. produced and purified the proteins, and performed the biochemical assays. Y.C., H. L., and L.Z. grew the crystals. Y.C., C.Y., Y.G., X.Y., and X.C. collected the X-ray data. Y. C., H.L., and J.G. determined the structures. Y.C., H.L., R.C., S.L., X.L., J.M., Z.H., J.L., and J.G. analyzed the data. Y.C., H.L., Z.H., J.L., and J.G. wrote the manuscript.

## Additional information

**Competing interests:** The authors declare no competing interests.

