## [Peer Review File · Nature Communications]

Reviewers' Comments:

Reviewer #1:

Remarks to the Author:

The structural and biochemical work on African Swine Fever Virus DNA ligase by Chen and coworkers shows the presence of a novel domain involved in DNA binding which replaces the DBD domain, present for example in the human DNA ligase. Biochemical results on mutants also presented confirm unsurprisingly its role. The second particularity of this enzyme is the tolerance towards base mismatches at the repair site which has also been addressed by biochemical data. The biochemical data confirm essentially the much more exhaustive study by Lamarche et al, 2005.

Despite the presence of 4 potentially exciting different high-resolution X-ray structures representing different states of ASFV DNA ligase (ATP bound open form in complex with nicked DNA, sealed and unsealed structure with a CT mismatch, unsealed structure with a CG base pair), the results are presented extremely inefficiently and it is difficult to extract any new information from the work presented.

Kinetic data and the effect of some mutants are supposed to support the story but finally do not contribute much to the understanding.

Some information about the dynamics of the process of DNA ligation could theoretically be extracted from the structural information on the different states (nick recognition, nick site adenylation on the 5' phosphate and sealing of the nick site by the nucleophilic attack of the 3' hydroxyl group) but the paper does not treat this very interesting dynamic aspect.

The central question about the structural base of the tolerance towards base mismatches at the nick site remains unanswered. The relevant structural superpositions and comparisons with ligases from other organisms are missing.

The bibliography is incomplete, in particular the work on the related human DNA ligase is neither sufficiently exploited nor cited such as the work by , Pascal et al, 2004, Ochi et al, 2010 and Cotner-Gohara et al., 2010, which also deal with dynamic aspects of ligase action.

In particular, the illustrations of the paper are extremely inefficient in supporting the two lines of argumentation.

In summary, the work of the authors made highly valuable structural information available, which has been exploited very poorly.

In this form the work is not suitable for publication in Nature Communications.

Some smaller issues are addressed below:

The introduction is approximate with commonplaces such as "one of the most complicated viruses known to date".

An introduction more in line with the wider audience of Nature Communications centered on fidelity and domain structure of DNA ligases in general would have been more adequate.

Electron density should be shown which supports the presence or the absence of the nick in the DNA substrate for the different structures.

The Rcryst of the data on the open conformation is unacceptably high and needs commenting.

The Materials and Methods section is much too detailed.

Reviewer #2:

Remarks to the Author:

This paper contains structural and biochemical analysis of the DNA ligase from a porcine virus. The structure has strong agricultural implications, as a target against this virus. The authors have done an extensive analysis. Although the work reads well overall, it is somewhat colloquial and has some grammar errors, and could be improved with a native English editor. Scientifically, the most interesting aspect of this work is this DNA ligase can work on certain mismatches, and this work

provides the structural rationale for how the mismatches are accommodated. Although the authors compare their protein qualitatively with the human protein, a quantitative analysis may reveal additional mechanisms. The second most interesting aspect is that the porcine virus has replaced the main DNA binding domain with its own. It is intriguing to consider what advantage this would have for the virus. In general the work appears interesting and worthwhile.

1. Figure 1A is difficult to read. I cannot see where CG and CT are on the left and can only see 7 of 10 lines on the right. On the right side, scaling to 60% will allow more separation. How many times was the time course done independently (e.g. different days)? (This should be given for all biochemistry figures). Was there any bias observed when fitting the curves for the ligation catalytic reaction?
2. What is the minimum font on the figures? Some seem smaller than 10 pt font.
3. What is the sequence conservation in the AD and OB domains between human and Asfv? Is there domain similarity to archaeal or human ligase (Mol Cell. 2006 Oct 20;24(2):279-91; Biochemistry. 2010 Jul 27;49(29):6165-76)? In Figure 1B, it would be helpful to include the correspondence of each domain to the residue number.
4. The ATP in Figure 1D should be shown with a simulated annealing omit map.
5. In Figure 2B, it would be good to have the ATP position somehow demarked.
6. Is Figure 2 a model or a crystal structure? My interpretation of the text is that the CT1 crystal domains were overlaid onto another crystal form. It would be better if refinement used the CT1 domains as a restraint and the figure showed the refined model corresponding to the crystal (not an overlay).
7. In Table 1, can you provide the overall Wilson B and the average B factor for protein, DNA, water, and ATP?
8. A depiction of the ligase domains' interactions (highlighting those conserved) with the DNA would be helpful. Programs such as DNAPRODB can create these automatically. How is the binding of each domain related to the major and minor grooves? It is interesting that the NTD of the AsfLigase is binding deep into the major groove, possibly similar to the DBD of human ligase. Is the number or relative proportion of bases that it's interacting with similar to the human protein? Do the interactions lead to any distortion in the DNA?
9. In line 191, the authors write "Compared to the OB domain, the AD domain forms many more interactions with the AsfvLIG NTD domain." What is the relative surface area, as calculated by a program such as PISA? From the figures, the area looks similar.
10. The biochemistry in Figure 4 shows that the NTD is essential for DNA binding. How does that compare to human ligase DBD? What is LD? It is not defined in text or figure legend.
11. The most interesting biological aspect of this structure is the ability to ligate mismatches. A faded gray outline underlay with the WC base pair would be helpful, if possible. Is there any selection for, neutral or against non-Watson Crick basepairing (e.g. Hoogsteen). Are there any compensating distortions in the neighboring basepairs.
12. The 402 and 403 mutants are interesting, as they seem to be stabilizing the mispair ligation. How do the mutant work on Watson Crick base pairs? Would you predict that they would have less effect on those than on mismatches?
13. It would be helpful to have the double, triple and quadruple mutants defined in the figure legend (and not just in the main text).
14. A figure showing the catalytic mechanism for ligation would provide more clarity on the impact of the changes in the porcine virus active site.
15. One interesting aspect of the porcine viral ligase is the replacement of the DBD. What advantage for the virus would the new DBD provide? The human protein interacts with PCNA. Does the virus encode PCNA? Does the virus contain a PIP to interact with the host PCNA?
16. Please have a native English speaker look through the text. These are the sentences that I noticed, but there may be more.
 - a. "Very surprise, no ligation activity was observed for the triple mutant even at high concentration"

- b. using Gel filtration buffer
- c. For catalysis assays, A 10- μ L reaction system
- d. Suggestion. Remove "very". It occurs multiple times in the text and its presence is more distracting than informative.

Response to the reviewers:

We sincerely thank both reviewers for reading our manuscript with great care, and we also thank the reviewers for their helpful comments, encouragement, and criticisms. Based on these comments and suggestions, we have carefully revised our manuscript with major changes highlighted in red in the revised manuscript. We believe that the quality of our manuscript has been significantly improved. The following are our point-to-point responses to the reviewers' comments.

Reviewers' comments:

Reviewer #1 (Remarks to the Author):

The structural and biochemical work on African Swine Fever Virus DNA ligase by Chen and coworkers shows the presence of a novel domain involved in DNA binding which replaces the DBD domain, present for example in the human DNA ligase. Biochemical results on mutants also presented confirm unsurprisingly its role. The second particularity of this enzyme is the tolerance towards base mismatches at the repair site which has also been addressed by biochemical data. The biochemical data confirm essentially the much more exhaustive study by Lamarche et al, 2005. Despite the presence of 4 potentially exciting different high-resolution X-ray structures representing different states of ASFV DNA ligase (ATP bound open form in complex with nicked DNA, sealed and unsealed structure with a CT mismatch, unsealed structure with a CG base pair), the results are presented extremely inefficiently and it is difficult to extract any new information from the work presented. Kinetic data and the effect of some mutants are supposed to support the story but finally do not contribute much to the understanding. Some information about the dynamics of the process of DNA ligation could theoretically be extracted from the structural information on the different states (nick recognition, nick site adenylation on the 5' phosphate and sealing of the nick site by the nucleophilic attack of the 3' hydroxyl group) but the paper does not treat this very interesting dynamic aspect. The central question about the structural base of the tolerance towards base mismatches at the nick site remains unanswered. The relevant structural superpositions and comparisons with ligases from other organisms are missing. The bibliography is incomplete, in particular the work on the related human DNA ligase is neither sufficiently exploited nor cited such as the work by, Pascal et al, 2004, Ochi et al, 2010 and Cotner-Gohara et al., 2010, which also deal with dynamic aspects of ligase action. In particular, the illustrations of the paper are extremely inefficient in supporting the two lines of argumentation. In summary, the work of the authors made highly valuable structural information available, which has been exploited very poorly. In this form the work is not suitable for publication in Nature Communications.

Response: We sincerely thank the reviewer for all the helpful comments, suggestions, and criticisms as well. We also thank the reviewer for confirming the importance and novelty of our structures. Based on these comments, we have carefully revised our manuscript. In addition to the structural comparison with the homologous protein, we also included many new catalytic assay results in the revised manuscript.

As revealed by structural analysis (Figs. 1C and S3A), AsfvLIG follows the same ATP binding and catalytic mechanism as the homologous proteins. The AsfvLIG NTD domain represents a novel DNA-binding fold and it mimics the DBD domains of the homology ligases in the catalytic form complex, which adopts a ring-like conformation. NTD and DBD cover a similar number of base pairs; however, in contrast to DBD, NTD does not form any strong interactions with the OB domain. Therefore, the interactions between substrate DNA and the other two domains of AsfvLIG become more important for the catalytic complex assembly.

Though they were not involved in the direct catalytic process, four nick site residues (Asn153 and Leu211 of the AD domain, and Leu402 and Gln403 of the OB domain) affect the catalytic efficiency of both Watson-Crick paired and mismatched DNA substrates (please see the figures 5, 6, S8, and S9). Compared to the corresponding residues, one Asp and one Arg (which form a salt bridge) and two Phe residues, the local conformation formed by the unique nick site residues of AsfvLIG is much more flexible. Instead of direct nick recognition, nick site adenylation on the 5'-phosphate and sealing of the nick site, our new ligation assay results indicated that the low fidelity of AsfvLIG is caused by the unique nick site residues, which are flexible and have evolved to accommodate various base pairs (both matched and mismatched). Compared to the other three residues, Gln403 appears to be more important for the catalytic efficiency of AsfvLIG, perhaps due to its ability in form an H-bond interaction with the nucleobase of the nick site nucleotides (please see Fig. 5B).

We also thank the reviewer for all the wonderful references, which have been discussed and cited in the revised manuscript.

Some smaller issues are addressed below:

The introduction is approximate with commonplaces such as “one of the most complicated viruses known to date”.

Response: Thanks for the helpful comment. The introduction section has been carefully rewritten in the revised manuscript.

An introduction more in line with the wider audience of Nature Communications centered on fidelity and domain structure of DNA ligases in general would have been more adequate.

Response: Thanks for the great suggestion. The introduction section has been rewritten in the revised manuscript, both fidelity and domain architecture of ligases were introduced.

Electron density should be shown which supports the presence or the absence of the nick in the DNA substrate for the different structures.

Response: Thanks for the suggestion. The electron density maps have been included in supplementary Fig. S7A-C, showing the presence or absence of the nick

in the three catalytic form structures. In the non-catalytic form structure, the nick DNA was bound by the NTD domains of two AsfvLIG molecules. Due to the potential strand switching and/or base shifting, the DNA electron density is averaged and no nick can be identified in the structure.

The Rcryst of the data on the open conformation is unacceptably high and needs commenting.

Response: Thanks for the comment. We examined our data collection and refinement statistic table (Table S5), and the high Rcrystal (24.7%) is for the catalytic form AsfvLIG:CT2 structure. Unlike the other three AsfvLIG structures, the AsfvLIG:CT2 structure has much higher solvent content (63%). In the crystal lattice, no symmetry-related molecule interacts with the NTD domain, though the residues interacting with DNA are clearly ordered (please see the left panel of the figure below), several loop regions that do not bind DNA are disordered in some content (supported by the weak $2F_o-F_c$ electron density shown below; the map was contoured at the 1.0 sigma level). Disorder of these NTD residues may contribute to the high Rcrystal value of the structure.

The AsfvLIG:CT2 crystals are very fragile; to obtain the diffraction data reported in the manuscript, hundreds of crystals were tested previously. Recently, we screened many more AsfvLIG:CT2 crystals, and most of them diffracted weakly (worse than 3.5Å). Though a couple crystals diffracted to 2.7-2.8 Å, refinement resulted in a similar Rcrystal value as the current structure. We then checked the Protein Database Bank (PDB) and compared the current structure to many structures with similar resolution (2.7Å); the Rcrystal of our structure is comparable. Therefore, we believe that our structure is acceptable and it can provide highly useful information, especially the conformation of the sealed C:T pair, which has been shown in Figs. 5B and S7C.

The Materials and Methods section is much too detailed.

Response: Thanks for the helpful comment. Some of the very detailed information has been deleted from the Materials and Methods section.

Reviewer #2 (Remarks to the Author):

This paper contains structural and biochemical analysis of the DNA ligase from a porcine virus. The structure has strong agricultural implications, as a target against this virus. The authors have done an extensive analysis. Although the work reads well overall, it is somewhat colloquial and has some grammar errors, and could be improved with a native English editor. Scientifically, the most interesting aspect of this work is this DNA ligase can work on certain mismatches, and this work provides the structural rationale for how the mismatches are accommodated. Although the authors compare their protein qualitatively with the human protein, a quantitative analysis may reveal additional mechanisms. The second most interesting aspect is that the porcine virus has replaced the main DNA binding domain with its own. It is intriguing to consider what advantage this would have for the virus. In general the work appears interesting and worthwhile.

Response: We sincerely thank the reviewer for all the useful comments and encouragement. As suggested by the reviewer, we have redone many of the in vitro ligation assays using a time-course and calculated the reaction rates for all the WT and mutant proteins. AsfvLIG has several unique residues at the nick site, including Asn153, Leu221, Leu402, and Gln403; compared to Asp570 and Arg871 (which form a salt bridge), and Phe536 and Phe872 of HsLIG1 and the corresponding residues in other homologous proteins, the nick side conformation of AsfvLIG is much more flexible. Besides mismatched substrates, our new ligation assays (please see the updated Figs. 5E-F, 6D-E, S8, and S9 in the revised manuscript) revealed that mutations of these nick site residues also significantly lowered the ligation rates of the Watson-Crick paired substrates. These observations suggested that, instead of discrimination, these flexible nick site residues of AsfvLIG are evolved to accommodate both mismatched and matched substrates, resulting in the apparent low fidelity of the protein.

Also, as suggested by the reviewer, our manuscript has been edited by a native English speaker from a language services company.

1. Figure 1A is difficult to read. I cannot see where CG and CT are on the left and can only see 7 of 10 lines on the right. On the right side, scaling to 60% will allow more separation. How many times was the time course done independently (e.g. different days)? (This should be given for all biochemistry figures). Was there any bias observed when fitting the curves for the ligation catalytic reaction?

Response: Thanks for the helpful comments and suggestions. The right panel of Fig. 1A has been scaled to 60% and it was renamed as Fig. 1B in the revised manuscript. All the experiments were repeated three times on different days. In addition to the mean values, the standard deviation (SD) values are also given for all the biochemistry figures.

2. What is the minimum font on the figures? Some seem smaller than 10 pt font.

Response: The minimum font is 10 pt on some figures. To make them more readable, we have increased the font size in the updated figures.

3. What is the sequence conservation in the AD and OB domains between human and Asfv? Is there domain similarity to archaeal or human ligase (Mol Cell. 2006 Oct 20;24(2):279-91; Biochemistry. 2010 Jul 27;49(29):6165-76)? In Figure 1B, it would be helpful to include the correspondence of each domain to the residue number.

Response: *We sincerely thank the reviewer for all the suggestions and wonderful references, which have been cited in the manuscript. AsfvLIG and HsLIG1 have some sequence conservation in their AD and OB domains; however, as depicted in the updated supplementary figure S1, the sequence conservation is very low. The sequence identity and similarity are about 10% and 20%, respectively, between the AD domains of the two proteins, and they are even lower for the OB domains, at only about 5% and 10%, respectively. Instead of AsfvLIG, the domain architectures of human ligases (please see the figure below) and archaeal ligases are more like each other; besides the AD and OB domains, the DBD domains are also highly conserved in the human and archaeal ligases. The residue numbers of each individual domains have been included in Figure 1B, which was renamed as Figure 1A in the revised manuscript.*

4. The ATP in Figure 1D should be shown with a simulated annealing omit map.

Response: *Done as suggested.*

5. In Figure 2B, it would be good to have the ATP position somehow demarked.

Response: *Thanks for the suggestion. For the HsLIG1 structure, the AMP that pyrophosphate linked to the 5'-P of the downstream DNA has been shown as red spheres in Figure 2B. Though one ATP was captured in the non-catalytic form AsfvLIG structure, it was not captured in the catalytic form structures. Therefore, we did not demark the ATP position in the AsfvLIG structure in Figure 2B.*

6. Is Figure 2 a model or a crystal structure? My interpretation of the text is that the CT1 crystal domains were overlaid onto another crystal form. It would be better if refinement used the CT1 domains as a restraint and the figure showed the refined model corresponding to the crystal (not an overlay).

Response: *Thanks for the helpful comments. Figure 2A is the real structure of the*

AsfvLIG:CT1 complex. To make it clearer, we have re-written the legend of Figure 2A in the revised manuscript.

7. In Table 1, can you provide the overall Wilson B and the average B factor for protein, DNA, water, and ATP?

Response: Done as suggested.

8. A depiction of the ligase domains' interactions (highlighting those conserved) with the DNA would be helpful. Programs such as DNAPRODB can create these automatically. How is the binding of each domain related to the major and minor grooves? It is interesting that the NTD of the AsfLigase is binding deep into the major groove, possibly similar to the DBD of human ligase. Is the number or relative proportion of bases that it's interacting with similar to the human protein? Do the interactions lead to any distortion in the DNA?

Response: We sincerely thank the reviewer for the great suggestion of this program. In the revised manuscript, we included two new figures, Fig. 3G and Fig. S4 (which is shown below), demonstrating the interaction between DNA and each individual domain of AsfvLIG. As depicted in Fig. S4B, the NTD and OB domains form 3 and 2 interactions with the DNA at the minor groove, respectively; though it is much bigger than the other two domains, the AD domain only forms one interaction with the DNA at the major groove. The sizes and overall folds of AsfvLIG NTD and HsLIG1 DBD are very different from each other, however, they covered a similar number of DNA base pairs in the catalytic form complexes (please see Fig. S5C). As revealed by the non-catalytic form AsfvLIG structure (Fig. 1C), interaction with NTD alone will not cause any distortion in the DNA; however, when the DNA is bound by all the three domains, it was bent in the nick site (Fig. S5B).

9. In line 191, the authors write “Compared to the OB domain, the AD domain forms many more interactions with the AsfvLIG NTD domain.” What is the relative surface area, as calculated by a program such as PISA? From the figures, the area looks similar.

Response: As calculated by the PISA program, the surface areas of the NTD, AD, and OB domains are 7239 Å², 9368 Å², and 5636 Å², respectively. The relative surface area between the NTD and OB domains is 157 Å², whereas it is 331 Å² between the NTD and AD domains.

10. The biochemistry in Figure 4 shows that the NTD is essential for DNA binding. How does that compare to human ligase DBD? What is LD? It is not defined in text or figure legend.

Response: As reported by Pascal and coworkers (Nature, 2004, 432:473-478), the DBD is important for DNA binding and catalysis of HsLIG1; deletion of DBD lowered the substrate binding affinity by >75-fold and reduced the catalytic efficiency of HsLIG1 by >4x10⁵-fold. As depicted in Figs. 4E and 4F, though the DNA binding ability of NTD is weaker than WT AsfvLIG, deletion of the NTD (for AsfvLIG ΔN) completely abolished the protein’s DNA binding and ligation activities. These observations suggested that, compared to HsLIG DBD, AsfvLIG NTD makes a similar (or even greater) contribution to the ligases’ DNA binding and ligation activity.

LD stands for AsfvLIG mutant with the NTD amino acids 85-92 replaced by two Gly residues. The corresponding figure has been moved to the supplementary section and renamed as Fig. S6E; LD was defined in both the main text and the figure legend.

11. The most interesting biological aspect of this structure is the ability to ligate mismatches. A faded gray outline underlay with the WC base pair would be helpful, if possible. Is there any selection for, neutral or against non-Watson Crick base pairing (e.g. Hoogsteen). Are there any compensating distortions in the neighboring base pairs?

Response: As shown in Fig. S2 and summarized in Table S2, AsfvLIG can catalyze the ligation reaction of various substrates. Among the non-Watson-Crick paired substrates, AsfvLIG prefers the substrates with a pyrimidine nucleotide at the template strands, such as DNA-TC, DNA-CT, DNA-TG, and DNA-CA. It can catalyze the ligation reaction of DNA-GG, but it dislikes the other three Hoogsteen-paired substrates (DNA-GA, DNA-AG, and DNA-AA).

The Watson-Crick C:G pair was not highlighted with a faded gray outline in the manuscript, but the detailed conformation of the C:G pair and its comparison with the C:T pair have been included in supplementary Fig. S7. The DNAs were bent in the catalytic form AsfvLIG complexes (Fig. S5B), however, the different nick site base pairs (C:G vs C:T) did not cause any additional compensating distortions in the neighboring base pairs (Fig. S7E).

12. The 402 and 403 mutants are interesting, as they seem to be stabilizing the mispair ligation. How do the mutant work on Watson Crick base pairs? Would you predict that they would have less effect on those than on mismatches?

Response: *We sincerely thank the reviewer for the wonderful suggestion. Besides DNA-CT and DNA-TC, we also performed in vitro ligation assays for the four Watson-Crick paired substrates; the results have been included in the revised manuscript (please see Figs. 5E-F, 6D, S8, and S9). Unexpectedly, the mutants with single (L402R or Q403F), double (L402R/Q403F), triple (N153D/L402R/Q403F), or quadruple (N153D/L211F/L402R/Q403F) mutations all showed much weaker ligation activity towards all tested substrates (both matched and mismatched). These observations strongly suggested that, instead of discrimination, the nick site residues have evolved to form a flexible conformation, which can accommodate various substrates, resulting in the apparent low fidelity of the protein.*

13. It would be helpful to have the double, triple and quadruple mutants defined in the figure legend (and not just in the main text).

Response: *Done as suggested.*

14. A figure showing the catalytic mechanism for ligation would provide more clarity on the impact of the changes in the porcine virus active site.

Response: *Thanks for the helpful suggestion. The catalytic mechanism of ATP-dependent ligase has been included in supplementary figure S3B in the revised manuscript.*

15. One interesting aspect of the porcine viral ligase is the replacement of the DBD. What advantage for the virus would the new DBD provide? The human protein interacts with PCNA. Does the virus encode PCNA? Does the virus contain a PIP to interact with the host PCNA?

Response: *In combination with the unique nick site residues, the DBD of AsfvLIG provides the protein with good tolerance in various base pairs; AsfvLIG, AsfvAP nuclease, and AsfvPolX (which is a low fidelity DNA polymerase) form one complete repair system, which can efficiently repair the DNA damage caused by the oxidative environment. In addition to the genome stability maintenance, this low fidelity repair system also plays an important role in quick genotype formation, which may provide the virus with a better ability of surviving under different conditions.*

Though it has not been experimentally verified, the virus encodes one protein (E301R) that shares weak sequence similarity with PCNA. However, the PIP-containing protein that can interact with E301R or host PCNA has not been reported.

16. Please have a native English speaker look through the text. These are the sentences that I noticed, but there may be more.

- a. "Very surprise, no ligation activity was observed for the triple mutant even at high concentration"
- b. using Gel filtration buffer
- c. For catalysis assays, A 10- μ L reaction system
- d. Suggestion. Remove "very". It occurs multiple times in the text and its presence is more distracting than informative.

Response: We sincerely thank the reviewer for the helpful suggestion and for reading our manuscript with great care. We have carefully revised our manuscript, which was also polished by a native English speaker from a language services company.

Reviewers' Comments:

Reviewer #1:

Remarks to the Author:

The answers of the authors address only partially the issues I raised.

I think, I was not explicit enough in my first review:

a) The mutational work done on Asfv ligase, in particular the choice of the mutations, is irrelevant for the explanation of the error tolerance of the enzyme. The mutagenesis does not tell anything about the importance of the residues for fidelity or tolerance to mismatches. The choice of the corresponding residues in the human structure does not really make sense. The human residues F872, D570, R871 and F635 form through a salt bridge and stacking interactions an interaction network as shown correctly in Fig 6C. The change of isolated residues is not expected to lead to any acquired function. In order to be conclusive, the mutational analysis needed to be done on the human enzyme showing that the disruption of the interactions lead to an increased tolerance for mismatches at the ligation site. The mutagenesis only shows that indeed these residues contacting the minor groove are important for the catalytic efficiency.

b) As AMP is absent it cannot be excluded that the "catalytic structures" are rather artifacts of the crystal packing which mimic pretty well the catalytic state. The results presented in Fig. 6E show clearly that a 2:1 complex incompatible with the crystal structure is possible in solution, what means that the presented structure only represents one of several possible structures where the OB domain may not contact the DNA. The manuscript states that the mutants affect dDNA binding, but in reality the mutants do not affect the affinity for DNA, they have only appear to favor dimerization of asfv ligase when it is bound to the substrate.

The contribution of the AD domain to affinity has not been studied, only the one of the N-terminal domain alone and the full length protein have been studied. The construct N-terminal domain + AD domain should also be analyzed. It is very well possible, that this construct has already the full affinity which is not increased by the presence of the OB domain.

c) An open form is also observed but hardly analyzed. By consequence the discussion is centered too much onto the "closed" form.

d) The positioning of the 5' phosphate and the 3'-OH in presence and in absence of a mismatch should be shown, compared and analyzed.

For these 4 points, the revised manuscript does not show any improvements.

The only sound evidence presented concerns the use by asfv ligase of a new type of N-terminal domain instead of the canonical N-terminal domains of DNA ligases and the contribution of a loop of the OB domain binding in the minor groove to catalytic efficiency which represents the exploitable part of the mutagenesis experiments.

In the first review, I have been hoping for a global strengthening of the message of the article and I did not go much in the details of the manuscript what I now do below:

The bibliography is still incomplete:

1. A separate reference should be given for the localization of the DNA replication, which is Rojo et al., 1999, but the contribution of the nucleus and the nature of the replication intermediates seems still to be up to debate. It should also be mentioned that the DNA ligase may also play a role in the ligation of Okazaki fragments occurring during replication.
2. The sentence "Owing to the weak diffraction, no ..." should be more explicit. What is "weak diffraction?"
3. How have the rmsd values been calculated? Based only on the superposition of C α atoms or of all backbone atoms. How many residues have been excluded from the calculation of the rmsd?
4. "One ATP was captured..." A bound ATP molecule is present in the non-catalytical structure.
5. "nick site composition" For example: The ligase residues in proximity of the nick in the substrate DNA.
6. Correct repairs "Unique features of ASFV repair enzymes",

7. One of the proposed modifications is problematic. In the section about the binding to nicked versus dsDNA:

"To clarify the function of AsfvLIG NTD, we carried out an in vitro DNA binding assay using nick and duplex DNA-CG. WT AsfvLIG can bind both nick and duplex DNA-CG (Figs. 4E and S6C); within the concentration range of 0.2-0.8 μ M, the nick DNA binding affinity of WT AsfvLIG is about 2-fold higher than that of duplex DNA. Compared to WT AsfvLIG, the DNA binding affinity of NTD is much weaker: at a 1.6 μ M concentration, NTD only binds about 20% nick DNA-CG and 9% DNA-CG duplex. Similar nick DNA preference was also observed for HsLIG1 and its DBD domain previously²²."

Looking at Fig. S6C the affinity difference for both substrate does not appear to be obvious and a statement that both substrates are bound with similar affinity would be sufficient.

8. "In fact, within the AD and OB domains, Leu402 and Gln403 are the only two residues projecting into the minor groove of the DNA (Fig. S4C)." This sentence is in contradiction with Fig. S4A, which shows also Y363 for a minor groove interaction.

9. "To further support this hypothesis, we carried out in vitro DNA binding assays (Figs. 6E-F)." These results are misinterpreted. The DNA binding affinity is not changed but some mutants favor dimerization of asfv ligase in a mode incompatible with the "catalytic" crystal structure.

10. The newly introduced diagram S4B needs explanation. What does H6, H9, S8 etc. mean?

11. It should be stated that the analysis using kobs is less complete than previous work by Lamarche who did a full enzymatic characterization leading to kcat's and Kd's.

12. The curves for the fraction of ligated substrate seem to level off and reach a plateau value different from 100 % ligation for weakly active mutants or substrates (Fig. 1B, Fig. 5EF, Fig. S9). Could you think of an explanation?

13. Fig. 1B, Fig. 6D, Table S2 the given kobs values are not plausible, from the curves you would expect maximal values in the order of $10e-1 \text{ min}^{-1}$, which is a factor 1000 bigger than the given numerical values. Most likely, the factor is $10e-3$ not $10e-6 \text{ min}^{-1}$.

14. Fig. 5. The order of the panel is non-standard.

15. Fig. S1: Giving the sequence of AsfarLig N-terminal domain in a panel A would be useful. It is a pity that its sequence only starts with the AD domain. The domains corresponding to the 2 different parts of the alignment should be indicated by bigger labels on top of panel A and B (which rather become panels B and C). The residue number for HsLIG1_AD on the left side of the top block is wrong.

16. Fig. S3B The catalytic residues such as the lysine should also be marked on the sequence of Fig. S1. There is an error in the scheme of the reaction. At the level of step 3 the link HO-P is missing. It would be better to write DNA and NH₂-LysLIG on one line for step 2.

17. Fig. S4A. The contacts of the AD domain should also be shown for the non-catalytic complex. S4B: Besides the non-explained symbols H9, S8 etc. suddenly some residues appear with a 3-letter code.

18. Fig. S6: Panel E should be split as it combines currently two different techniques, EMSA and ligation assay.

19. Fig. S7: The interacting water of Fig. 5A should be shown in panel E and F. A homogenous coloring scheme for the central base between panels A-C and panels D-F would have been nice. Also the open CG and CT complexes should have been superposed around the 5'-phosphate and the 3'-OH group. A very similar position of the two structures around the nick would explain why the ligation reaction is evenly efficient for base pairs or CG mismatches.

20. Table S1: I guess that the FAM label is linked to the 5'-phosphate as in the commercial 6FAM modification and not to the base, as drawn. Lines indicating base pairs are slightly offset. Do not draw continuous but dotted lines at the positions of the mismatches what applies also to Table S2.

21. Table S2: Uncertainties shall only be given with two significant decimals, which also determine the last digit of the result which is presented.

Reviewer #2:

Remarks to the Author:

Re: Crystal structures and implications of the error-prone DNA ligase of African swine fever virus.

The authors have improved the writing and added in more kinetic and structural analyses. Unfortunately, it is still not possible to assess the quality of the data, given the current figures (see below). Additionally, the greatest insight for how the viral ligase can ligate with mismatches, in comparison to the human ligases, remains obscure as currently reported.

- 1) Based on the figures provided, it is impossible to assess the quality of the time courses. For a proper determination of the activity, the rate must be linear over time. Many proteins display a burst of activity followed by a slower rate, indicative of product release being rate limiting. To measure k_{cat} , the initial linear rate should be measured. In the current figures, the data points are crowded at the bottom of the graph, making it impossible to view the linearity of the enzymatic activity or to distinguish between different mutants. It is not clear why the authors chose their Y axes as so high, given that the highest point in the data is $\sim 25\%$ of the Y axis maximum. In the one figure where the highest point goes to 50% of the y axis (Fig. S9), the authors have drawn their line over the entire course of the enzymatic reaction, drawing through both burst and the slower rate. Recommend that the authors match their y-axis to the maximal data and to only measure the initial linear region of the graph. At least three points should be used to draw the lines. Data points should be randomly on either side of the line. The data point positions should not be biased, with initial and final points below the line and middle points above the line, as observed in Fig. S9.
- 2) The major insight coming from this work is the allowance of the mismatch being ligated. The authors propose that the viral active site is more flexible. When one considers flexibility in a structure, it is standardly considered that there are protein regions that adopt multiple conformations. Is this what the authors intended? Based on Figure S7, it appears that the DNA is flexible, with a lack of base pairing. A CT base should have two H-bonds, but it appears to have only one, suggesting strain. The position of the phosphodiester backbone appears to be maintained. Does the phosphodiester backbone overlay, when overlaying the protein chain for the GC and the CT complexes? If that is the case, are there more phosphodiester stabilizing residues in the AsfvLig than in Lig1, 3, or 4? What is missing in the AsfvLig from Lig 1, 3, and/or 4 that ensures W-C fidelity? Could AsfvLig work on single stranded RNA or DNA?
- 3) The authors included a catalytic reaction schematic, as requested. However, it is a simple one that could be used in a biochemistry paper without the structure. It would be more helpful and inclusive of the results being reported if that schematic includes the active site residues and geometry, as this is a structure paper where the active site geometry is known.
- 4) The conservation between AD and OBD domains in porcine and human DNA ligases should be provided in text. It is interesting that despite the lower conservation, the structure is conserved.
- 5) The reference to the Ligase IV structure was not added. Structural comparison to Ligase III and Ligase IV were not done, which could provide information on the what makes the porcine virus enzyme unique. As above in #2, what is different from AsfvLig and the other Lig structures that allows for non-WC specificity? A part of this question is knowing what in the other lig structures enforces WC specificity.
- 6) Please include PDB ID for structures reported in this paper. It is helpful to the readers, if they wanted to look at them in a graphics program.

We sincerely thank both reviewers for reading our manuscript with great care, and for their helpful comments, encouragement, and criticisms. Based on these comments and suggestions, we have carefully revised our manuscript with major changes highlighted in red in the revised manuscript. The following are our point-to-point responses to the reviewers' comments.

Reviewers' comments:

Reviewer #1 (Remarks to the Author):

The answers of the authors address only partially the issues I raised.

I think, I was not explicit enough in my first review:

a) The mutational work done on *Asfv* ligase, in particular the choice of the mutations, is irrelevant for the explanation of the error tolerance of the enzyme. The mutagenesis does not tell anything about the importance of the residues for fidelity or tolerance to mismatches. The choice of the corresponding residues in the human structure does not really make sense. The human residues F872, D570, R871 and F635 form through a salt bridge and stacking interactions an interaction network as shown correctly in Fig 6C. The change of isolated residues is not expected to lead to any acquired function. In order to be conclusive, the mutational analysis needed to be done on the human enzyme showing that the disruption of the interactions lead to an increased tolerance for mismatches at the ligation site. The mutagenesis only shows that indeed these residues contacting the minor groove are important for the catalytic efficiency.

Response: We sincerely thank the reviewer for the very patient explanation and are very sorry for misunderstanding the reviewer's previous comments regarding the mutagenesis. Based on the reviewer's suggestions, we have constructed and purified three human *LIG1* proteins (aa 262-919), including the wild-type (WT) *HsLIG1*, R871L/F872Q double mutant (*HsLIG1-d*), and D570N/F635L/R871F/F872Q quadruple mutant (*HsLIG1-q*). In *HsLIG1-d* and *HsLIG1-q*, two and four ligase residues in proximity of the nick in the substrate DNA are replaced by the corresponding residues of *AsfvLIG1*. The gel-filtration profile and SDS-gel analysis results are depicted in the figure below.

(A) The typical gel-filtration profile of WT and mutant *HsLIG1* proteins. (B) SDS-gel analysis showing the purity of the *HsLIG1* proteins.

In consistent with previous study, our *in vitro* catalytic assays showed that WT HsLIG1 (Fig. S11) has very high ligation activity and high fidelity toward DNA substrates. Interestingly, besides the non-Watson-Crick paired DNA substrates, the Watson-Crick paired DNA ligation activity of HsLIG1-d was also significantly lowered (Fig. S12). And no clear ligation activity was observed for any DNA substrate for the HsLIG1-q mutant (Fig. S13). These observations clearly indicated that, in addition to fidelity, the four ligase residues (D570, F635, R871, and F872) also play critical role in the catalytic efficiency of HsLIG1. The related results are summarized in the figure below and in Fig.7 of the revised manuscript. In combination with the mutagenesis and *in vitro* catalytic assays (Figs. 6D, S2, S9, and S10) of AsfvLIG, our results clearly indicated that the ligase residues in proximity of the nick in DNA substrate could not be switched between AsfvLIG and HsLIG1.

Like all the characterized ATP-dependent DNA ligases, AsfvLIG catalyzes phosphodiester bond formation between adjacent 3'-OH and 5'-phosphate in DNA duplex through a similar three-step mechanism (Please see the Figure below and Fig. 8A in the revised manuscript). Conceptually, differentiation between matched and mismatched base pairs could take place in step 2 and/or step 3 of the reaction. Indicated by the concomitant accumulation of the adenylated DNA intermediates for the mismatches only, HsLig3 and Vaccinia virus DNA ligase attain part fidelity during step 3. However, no adenylated intermediates for either mismatched or

matched DNAs were detected during the *in vitro* DNA ligation assays catalyzed by *AsfvLIG* (Fig. S2) and *HsLig1* (Fig. S11). These observations indicate that once adenylation of the nick occurs, the subsequent nick closure step will quickly finish. In other words, these observations also indicate that *HsLig1* and *AsfvLIG* do not actively discriminate between match and mismatch during step 3; the discrimination or tolerance of mismatched DNAs solely occurs in step 2.

In addition to *in vitro* catalysis, we also compared the *in vitro* DNA binding behaviors of *AsfvLIG* and *HsLig1* (Please see the figure below and Figs. 8C-D in the revised manuscript). Similar to the four Watson-Crick paired DNAs, DNAs with mismatched base pairs at the 3'-end of the nick can all be efficiently bound by *AsfvLIG*; the binding affinities between *AsfvLIG* and all DNAs are very similar. Compared to *AsfvLIG*, *HsLig1* has similar binding affinity to mismatched DNAs with pyrimidines (C or T) on the template strand; however, when the protein concentrations are within the range of 0.2-0.8 μM , the binding affinities between *HsLig1* and mismatched DNAs with purines (A or G) on the template strands are significantly weaker than those of *AsfvLIG*. Very surprising, compared to the mismatched DNAs, the binding affinities between *HsLig1* and the four matched DNAs are much weaker.

In consistent with previous study (Showalter et al., 2006, Chemical Review, 106: 340-360), the obvious different DNA binding behavior, unique fold of the NTD domain, and unique residues at the nick site all confirm that *AsfvLIG* is one atypical DNA ligase. Before any *AsfvLIG* structure was reported, Showalter and coworkers

already demonstrated that *AsfvLIG* has very low adenylation activity towards DNAs with 3'-dideoxy- or 3'-amino-terminated nicks, compared to regular nick DNAs; these observations indicated that 3'-OH of the nick is a critical component of the active site architecture during 5'-P adenylation. We believe that the unique NTD domain and the unique residues at the nick site all contribute to the low fidelity of *AsfvLIG*. The extensive interactions between DNA and the NTD domain offer *AsfvLIG* with strong binding affinity to all matched and mismatched DNAs. In addition to matched DNAs, the strong binding also gives *AsfvLIG* enough opportunity to catalyze the ligation of certain mismatched DNAs. In incorporation with the NTD domain, the four nick site residues will help the reorientation of DNA 3'-OH and facilitate the ligation reaction. Compared to other nick site residues, Gln403 appears to be more important for the catalytic efficiency of *AsfvLIG*. As exemplified by the *AsfvLIG*:CT2 structure (Fig. 5B), Gln403 can form H-bond interaction with the mismatched C:T pair at nick 3'-end; in addition to catalytic efficiency, such interaction definitely also contributes to the low fidelity of *AsfvLIG*.

Different from *AsfvLIG*, *HsLig1* is a high fidelity DNA ligase. Though *HsLig1* also discriminates between match and mismatch during step 2 only, our *in vitro* binding assays showed (Fig. 8D) that *HsLig1* has much weaker binding affinities towards the matched nick DNAs, compared to the mismatched nick DNAs. These observations clearly indicated that the active site architecture will undergo certain conformational change during the adenylation process of DNA 5'-P. To ensure the high fidelity of *HsLig1*, the matched DNAs must be much more efficient during this reposition process, compared to the mismatched nick DNAs. Based on the only available *HsLig1*:DNA complex structure, it has been previously proposed that *HsLig1* can impose some local distortion on the duplex, resulting in the 3'-OH and the adenylated 5'-P in positions appropriate for nick sealing. However, more structures, especially the structures of *HsLig1* complexed with mismatched DNAs, are required to fully understand the basis for discrimination between matched and mismatched DNAs during the 3'-OH and 5'-P reposition process.

b) As AMP is absent it cannot be excluded that the “catalytic structures” are rather artifacts of the crystal packing which mimic pretty well the catalytic state. The results presented in Fig. 6E show clearly that a 2:1 complex incompatible with the crystal structure is possible in solution, what means that the presented structure only represents one of several possible structures where the OB domain may not contact the DNA. The manuscript states that the mutants affect dsDNA binding, but in reality the mutants do not affect the affinity for DNA, they have only appear to favor dimerization of *asfv* ligase when it is bound to the substrate. The contribution of the AD domain to affinity has not been studied, only the one of the N-terminal domain alone and the full length protein have been studied. The construct N-terminal domain + AD domain should also be analyzed. It is very well possible, that this construct has already the full affinity which is not increased by the presence of the OB domain.

Response: We totally agree with the reviewer that no AMP was captured in our “catalytic structure”. The three catalytic structures belong to two different space

groups: P2₁2₁2₁ for *AsfvLIG*:CT2 and P2₁ for both *AsfvLIG*:CG and *AsfvLIG*:CT1. The molecular packing of *AsfvLIG*:CT2 is very different from the other two structures in the crystal lattice. However, as indicated by the low root mean square deviations (rmsd, about 0.75 Å, based on the superposition of 404 pairs of C α atoms) among them, the three complex structures are very similar to each other. Besides *HsLig1*:DNA complex (Figs. 2C and S4B), we also compared our *AsfvLIG*:CT1 structure with the catalytic form *HsLIG3*:DNA complex (Fig. S9A), showing that these two structures have similar overall folds. Based on these observations, we believe that our structures represent the real catalytic form structure and they can provide useful information, such as DNA substrate binding, catalytic complex assembly, and interaction between the NTD domain and the other two domains of *AsfvLIG*.

We also thank the reviewer for the helpful comments regarding the binding affinity and binding state between *AsfvLIG* and DNA substrates. We have carefully rewritten the statements in the revised manuscript. Based on the reviewer's comments, we constructed and purified one *AsfvLIG* protein with the OB domain deleted (referred as Δ OB). Δ OB can bind nick DNA and duplex DNA. As depicted in the figure below, the DNA-binding affinity of Δ OB is stronger than that of NTD domain of *AsfvLIG*. However, compared to the full-length WT *AsfvLIG*, the DNA-binding affinity of Δ OB is much weaker.

(A) and (B) *In vitro* DNA binding by WT and NTD of *AsfvLIG* and Δ OB of *AsfvLIG*, respectively.

(C) *In vitro* DNA ligation catalyzed by Δ OB of *AsfvLIG*.

As mentioned by the reviewer, the L402R or Q403F mutation within the OB domain will favor the dimerization of *AsfvLIG* when it is bound to the substrate. In addition to the 1:1 complex, we also noticed a significant amount of bands corresponding to the complex with Δ OB:DNA molar ratio of 2:1 during the *in vitro* DNA binding assay. To test whether these complexes are compatible with the catalysis, we also carried out *in vitro* ligation assay. As depicted in the figure above, Δ OB could not catalyze the DNA ligation reaction, suggesting that the OB domain is important for the catalytic activity of *AsfvLIG*. In combination with the DNA binding and ligation of WT *AsfvLIG* and NTD, our studies indicate that all the three domains (NTD, AD, and OB) of *AsfvLIG* are necessary for the correct catalytic form complex assembly and catalysis of the protein. The results of *in vitro* DNA binding and ligation assays in the presence of Δ OB have also been included in Figs. S7 in the revised manuscript.

c) An open form is also observed but hardly analyzed. By consequence the discussion is centered too much onto the “closed” form.

Response: We thank the reviewer for the helpful comments. The “open” form structure provides detailed information on ATP binding and it captured the complex with *AsfvLIG*:DNA molar ratio of 2:1, which is incompatible with the catalysis. In combination with the “closed” form structures, the “open” form structure also revealed some domain arrangement during the substrate binding. We have reflected all these key points in the revised manuscript. In terms of assembly of the catalytic complex and the detailed interaction between *AsfvLIG* and DNA, the “close” form structures provide much more valuable information. Compared to the “open” form structure (2.55 Å), the resolution (2.35 Å) of the “closed” form *AsfvLIG*:CT1 structure is also slightly higher, therefore, we mainly focus on the “closed” form structure during structural analysis.

d) The positioning of the 5' phosphate and the 3'-OH in presence and in absence of a mismatch should be shown, compared and analyzed.

Response: The positioning of the 5' phosphate and 3'-OH is similar in presence and in absence of mismatch. We have included one new panel (Fig. 5C) and discussed about the positioning of the 5' phosphate and 3'-OH in the revised manuscript.

For these 4 points, the revised manuscript does not show any improvements. The only sound evidence presented concerns the use by *asfv* ligase of a new type of N-terminal domain instead of the canonical N-terminal domains of DNA ligases and the contribution of a loop of the OB domain binding in the minor groove to catalytic efficiency which represents the exploitable part of the mutagenesis experiments. In the first review, I have been hoping for a global strengthening of the message of the article and I did not go much in the details of the manuscript what I now do below:

1. The bibliography is still incomplete. A separate reference should be given for the

localization of the DNA replication, which is Rojo et al., 1999, but the contribution of the nucleus and the nature of the replication intermediates seems still to be up to debate. It should also be mentioned that the DNA ligase may also play a role in the ligation of Okazaki fragments occurring during replication.

Response: We sincerely thank the reviewer for the helpful comments and wonderful reference, which (reference #8) has been included in the revised manuscript. We also mentioned the involvement of DNA ligase in the ligation of Okazaki fragments in the introduction section.

2. The sentence “Owing to the weak diffraction, no ...” should be more explicit. What is “weak diffraction?”

Response: Thanks for the helpful comment. In addition to AsfvLIG-DNA complexes, we also screened the sample of apo-AsfvLIG in this study. Though we got some apo-AsfvLIG crystals, they did not diffract well; the typical resolutions are around 8-10 Å. The term “weak diffraction” was used to indicate the low resolution of the diffraction data in previous manuscript. To avoid the possible confusion, we have deleted this term in the revised manuscript.

3. How have the rmsd values been calculated? Based only on the superposition of C-alpha atoms or of all backbone atoms. How many residues have been excluded from the calculation of the rmsd?

Response: The rmsd values are calculated based on the superposition of C-alpha atoms. In the revised manuscript, we compared AsfvLIG structure with all the three HsLIG structures, including HsLIG1, HsLIG3, and HsLIG4. In each case, there are different numbers of residues excluded from the AsfvLIG and HsLIG structure during rmsd calculation. Instead of the number of the excluded residues, we prefer to give the number of residues used for structural superposition, to keep the manuscript more concise. The detailed number has been given in the revised manuscript.

4. “One ATP was captured...” A bound ATP molecule is present in the non-catalytical structure.

Response: Thanks for the wonderful suggestion. We have replaced the sentence with “A bound ATP molecule is present in the non-catalytical structure” in the revised manuscript.

5. “nick site composition” For example: The ligase residues in proximity of the nick in the substrate DNA.

Response: Thanks for the wonderful suggestion. We have replaced the term with “The ligase residues in proximity of the nick in the substrate DNA” in the revised manuscript.

6. Correct repairs “Unique features of ASFV repair enzymes”.

Response: Thanks for the correction. We have fixed the mistake in the revised

manuscript.

7. One of the proposed modifications is problematic. In the section about the binding to nicked versus dsDNA:

“To clarify the function of AsfvLIG NTD, we carried out an in vitro DNA binding assay using nick and duplex DNA-CG. WT AsfvLIG can bind both nick and duplex DNA-CG (Figs. 4E and S6C); within the concentration range of 0.2-0.8 μ M, the nick DNA binding affinity of WT AsfvLIG is about 2-fold higher than that of duplex DNA. Compared to WT AsfvLIG, the DNA binding affinity of NTD is much weaker: at a 1.6 μ M concentration, NTD only binds about 20% nick DNA-CG and 9% DNA-CG duplex. Similar nick DNA preference was also observed for HsLIG1 and its DBD domain previously²².”

Looking at Fig. S6C the affinity difference for both substrate does not appear to be obvious and a statement that both substrates are bound with similar affinity would be sufficient.

Response: We have repeated the DNA binding assay for several more times. In consistent with our previous assays, the new results also suggest that WT AsfvLIG and NTD have some weak preference for the nick DNA. We thank the reviewer for the helpful comments and nice suggestion. In the revised manuscript, we have toned down the statements. Fig. S6C has been renumbered as Fig. S5C in the revised manuscript.

8. “In fact, within the AD and OB domains, Leu402 and Gln403 are the only two residues projecting into the minor groove of the DNA (Fig. S4C).” This sentence is in contradiction with Fig. S4A, which shows also Y363 form a minor groove interaction.

Response: We sincerely thank the reviewer for pointing our mistakes. Fig. S4A has been renumbered as Fig. S3A in the revised manuscript. By carefully examining the structures and Fig. S3A, we found that Leu211, Ala215, and Asn219 of AD domain and Tyr363, Leu402, and Gln403 of OB domain all interact with the DNA in the minor groove. Among them, Leu402 and 403 are the only two reside next to the nick. We have modified the sentence in the revised manuscript.

9. “To further support this hypothesis, we carried out in vitro DNA binding assays (Fig. 6E).” These results are misinterpreted. The DNA binding affinity is not changed but some mutants favor dimerization of asfv ligase in a mode incompatible with the “catalytic” crystal structure.

Response: Thanks for the correction. We have fixed the mistake in the revised manuscript.

10. The newly introduced diagram S4B needs explanation. What does H6, H9, S8 etc. mean?

Response: Fig. S4B has been renumbered as Fig. S3B in the revised manuscript. H6, H9, S8 etc represent the helices α 6 and α 9, and strand β 8 etc of AsfvLIG. The labels and legend of the figure have been updated.

11. It should be stated that the analysis using k_{obs} is less complete than previous work by Lamarche who did a full enzymatic characterization leading to k_{cat} 's and K_d 's.

Response: Thanks for the helpful comment. We have reflected this point and cited the reference in the first paragraph of the results section of the revised manuscript.

12. The curves for the fraction of ligated substrate seem to level off and reach a plateau value different from 100 % ligation for weakly active mutants or substrates (Fig. 1B, Fig. 5EF, Fig. S9). Could you think of an explanation?

Response: We are not sure about the exact basis underlying these phenomena, but we think that slow product releasing could be one potential cause of these phenomena.

13. Fig. 1B, Fig. 6D, Table S2 the given k_{obs} values are not plausible, from the curves you would expect maximal values in the order of $10e^{-1} \text{ min}^{-1}$, which is a factor 1000 bigger than the given numerical values. Most likely, the factor is $10e^{-3}$ not $10e^{-6} \text{ min}^{-1}$.

Response: We thank the reviewer for the helpful comments. We have carefully reanalyzed our data, which supported our previous results. Of course, we agree with the reviewer, some K_{obs} values can be written using the factor of $10e^{-3} \text{ min}^{-1}$, especially for the double, triple, and quadruple mutants of AsfvLIG. However, to keep the factor consistent, we prefer to use the factor of $10e^{-6} \text{ min}^{-1}$ for all WT and mutants of AsfvLIG.

14. Fig. 5. The order of the panel is non-standard.

Response: The panels in Fig. 5 have been rearranged according to the standard mode.

15. Fig. S1: Giving the sequence of AsfvLig N-terminal domain in a panel A would be useful. It is a pity that its sequence only starts with the AD domain. The domains corresponding to the 2 different parts of the alignment should be indicated by bigger labels on top of panel A and B (which rather become panels B and C). The residue number for HsLIG1_AD on the left side of the top block is wrong.

Response: We thank the reviewer for the very helpful comments. As suggested by the reviewer, a new panel (Fig. S1A) has been included in the figure, showing the sequence of AsfvLIG NTD. The NTD, AD domain, and OB domain of the proteins are labelled on the right side of the panels A, B, and C, respectively. We also thank the reviewer for pointing out our mistake on the residue number of HsLIG1_AD, which has been fixed.

16. Fig. S3B: The catalytic residues such as the lysine should also be marked on the sequence of Fig. S1. There is an error in the scheme of the reaction. At the level of step 3 the link HO-P is missing. It would be better to write DNA and NH₂-LysLIG on one line for step 2.

Response: We thank the reviewer for the very helpful comments. Fig. S3B has been moved to Fig. 8A in the revised manuscript. In addition to the catalytic lysine, we

also marked the other catalytic residues, such as Glu203 and Glu291, with red asterisks in the figure. The ligase residues in proximity of the nick in the substrate DNA are marked with green asterisks.

Thanks for pointing out our mistake in the scheme of the reaction, which has been fixed. As suggested by the reviewer, DNA and NH₂-LysLIG are written on one line for step 2.

17. Fig. S4A. The contacts of the AD domain should also be shown for the non-catalytic complex. S4B: Besides the non-explained symbols H9, S8 etc. suddenly some residues appear with a 3-letter code.

Response: We thank the reviewer for the very helpful comments. Fig. S4A has been renumbered as Fig. S3A in the revised manuscript. Unlike the catalytic form complex, in which DNA interacts with all the NTD, AD, and OB domains of *AsfvLIG*, the AD and OB domains do not interact with DNA in the non-catalytic form complex. Therefore, we did not include the non-catalytic complex in Fig. S3A.

H6, H9, S6, and S8 represent the helices 6 and 9, and strands 6 and 8 of *AsfvLIG*. The labels and legend of Fig. S4B have been updated in the revised manuscript. The 3-letter codes are also replaced by 1-letter codes in the updated Fig. S3B.

18. Fig. S6: Panel E should be split as it combines currently two different techniques, EMSA and ligation assay.

Response: Thanks for the suggestion. Fig. S6 has been renumbered as Fig. S5 in the revised manuscript. We have split the two panels in Fig. S5E and moved the EMSA assay result to Fig. S5D.

19. Fig. S7: The interacting water of Fig. 5A should be shown in panel E and F. A homogenous coloring scheme for the central base between panels A-C and panels D-F would have been nice. Also the open CG and CT complexes should have been superposed around the 5'-phosphate and the 3'-OH group. A very similar position of the two structures around the nick would explain why the ligation reaction is evenly efficient for base pairs or CT mismatches.

Response: We sincerely thank the reviewer for the wonderful suggestions. The interacting water of Fig. 5A has been shown in Fig. S6E and S6F. A homogenous coloring scheme has been used for the central base between panels A-C and panels E-F. The original Fig. S7D has been moved to Fig. 5 in the revised manuscript.

20. Table S1: I guess that the FAM label is linked to the 5'-phosphate as in the commercial 6FAM modification and not to the base, as drawn. Lines indicating base pairs are slightly offset. Do not draw continuous but dotted lines at the positions of the mismatches what applies also to Table S4.

Response: We thank the reviewer for the helpful comments and suggestions. We have fixed the mistakes in both Table S1 and S5, corresponding to the original Table S4.

21. Table S2: Uncertainties shall only be given with two significant decimals, which also determine the last digit of the result which is presented.

Response: Thanks for the suggestion. Two significant decimals are given for all the K_{obs} values in Table S2 and the text throughout the manuscript.

Reviewer #2 (Remarks to the Author):

Re: Crystal structures and implications of the error-prone DNA ligase of African swine fever virus.

The authors have improved the writing and added in more kinetic and structural analyses. Unfortunately, it is still not possible to assess the quality of the data, given the current figures (see below). Additionally, the greatest insight for how the viral ligase can ligate with mismatches, in comparison to the human ligases, remains obscure as currently reported.

Response: We sincerely thank the reviewer for the helpful comments. As suggested by both reviewers, we have compared *AsfvLIG* and all the three *HsLIG* structures in the revised manuscript (Figs. S1, S4, and S9). Though the AD and OB domains are conserved in all the four proteins, *AsfvLIG* possesses four very unique ligase residues (Asn153, Leu211, L402, and Q403) in the proximity of the nick in the substrate DNA. Previous, we showed that replacing these residues by the corresponding residues of *HsLIGs* significantly lowered *AsfvLIG*'s DNA ligation activity. As suggested by reviewer 1, we have constructed and purified wild type and two mutants of *HsLIG1*, in which the nick site residues are replaced by the corresponding residues of *AsfvLIG*. Similar to the *AsfvLIG* mutation, our *in vitro* ligation assays showed that mutation of the nick residues of *HsLIG1* also significantly impaired the protein's DNA ligating activity (Figs. 7 and S11-S13). These results clearly indicated that the nick site residues could not be switched between *AsfvLIG* and *HsLIG1*.

Like all the characterized ATP-dependent DNA ligases, *AsfvLIG* catalyzes phosphodiester bond formation between adjacent 3'-OH and 5'-phosphate in DNA duplex through a similar three-step mechanism. Conceptually, differentiation between matched and mismatched base pairs could take place in step 2 and/or step 3 of the reaction. Indicated by the concomitant accumulation of the adenylated DNA intermediates for the mismatches only, *HsLig3* and *Vaccinia virus* DNA ligase attain part fidelity during step 3. However, no adenylated intermediates for either mismatched or matched DNAs were detected during the *in vitro* DNA ligation assays catalyzed by *AsfvLIG* (Fig. S2) and *HsLig1* (Fig. S11). These observations indicate that once adenylation of the nick occurs, the subsequent nick closure step will quickly finish. In other words, these observations also indicate that *HsLig1* and *AsfvLIG* do not actively discriminate between match and mismatch during step 3; the discrimination or tolerance of mismatched DNAs solely occurs in step 2.

In addition to *in vitro* catalysis, we also compared the *in vitro* DNA binding behaviors of *AsfvLIG* and *HsLig1* (Figs. 8C-D). Similar to the four Watson-Crick paired DNAs, DNAs with mismatched base pairs at the 3'-end of the nick can all be efficiently bound by *AsfvLIG*; the binding affinities between *AsfvLIG* and all DNAs are very similar. Compared to *AsfvLIG*, *HsLig1* has similar binding affinity to mismatched DNAs with pyrimidines (C or T) on the template strand; however, when the protein concentrations are within the range of 0.2-0.8 μM , the binding affinities between *HsLig1* and mismatched DNAs with purines (A or G) on the template

strands are significantly weaker than those of *AsfvLIG*. Very surprising, compared to the mismatched DNAs, the binding affinities between *HsLig1* and the four matched DNAs are much weaker.

In consistent with previous study (Showalter et al., 2006, Chemical Review, 106: 340-360), the obvious different DNA binding behavior, unique fold of the NTD domain, and unique residues at the nick site all confirm that *AsfvLIG* is one atypical DNA ligase. Before any *AsfvLIG* structure was reported, Showalter and coworkers already demonstrated that *AsfvLIG* has very low adenylation activity towards DNAs with 3'-dideoxy- or 3'-amino-terminated nicks, compared to regular nick DNAs; these observations indicated that 3'-OH of the nick is a critical component of the active site architecture during 5'-P adenylation. We believe that the unique NTD domain and the unique residues at the nick site all contribute to the low fidelity of *AsfvLIG*. The extensive interactions between DNA and the NTD domain offer *AsfvLIG* with strong binding affinity to all matched and mismatched DNAs. In addition to matched DNAs, the strong binding also gives *AsfvLIG* enough opportunity to catalyze the ligation of certain mismatched DNAs. In incorporation with the NTD domain, the four nick site residues will help the reorientation of DNA 3'-OH and facilitate the ligation reaction. Compared to other nick site residues, Gln403 appears to be more important for the catalytic efficiency of *AsfvLIG*. As exemplified by the *AsfvLIG*:CT2 structure (Fig. 5B), Gln403 can form H-bond interaction with the mismatched C:T pair at nick 3'-end; in addition to catalytic efficiency, such interaction definitely also contributes to the low fidelity of *AsfvLIG*.

Different from *AsfvLIG*, *HsLig1* is a typical, high fidelity DNA ligase. Though *HsLig1* also discriminates between match and mismatch during step 2 only, our *in vitro* binding assays showed (Fig. 8D) that *HsLig1* has much weaker binding affinities towards the matched nick DNAs, compared to the mismatched nick DNAs. These observations clearly indicated that the active site architecture will undergo certain conformational change during the adenylation process of DNA 5'-P. To ensure the high fidelity of *HsLig1*, the matched DNAs must be much more efficient during this reposition process, compared to the mismatched nick DNAs. Based on the only available *HsLig1*:DNA complex structure, it has been previously proposed that *HsLig1* can impose some local distortion on the duplex, resulting in the 3'-OH and the adenylated 5'-P in positions appropriate for nick sealing. However, more structures, especially the structures of *HsLig1* complexed with mismatched DNAs, are required to fully understand the basis for discrimination between matched and mismatched DNAs during the 3'-OH and 5'-P reposition process.

1) Based on the figures provided, it is impossible to assess the quality of the time courses. For a proper determination of the activity, the rate must be linear over time. Many proteins display a burst of activity followed by a slower rate, indicative of product release being rate limiting. To measure k_{cat} , the initial linear rate should be measured. In the current figures, the data points are crowded at the bottom of the graph, making it impossible to view the linearity of the enzymatic activity or to distinguish between different mutants. It is not clear why the authors chose their Y axes as so high, given that the highest point in the data is

~25% of the Y axis maximum. In the one figure where the highest point goes to 50% of the y axis (Fig. S9), the authors have drawn their line over the entire course of the enzymatic reaction, drawing through both burst and the slower rate. Recommend that the authors match their y-axis to the maximal data and to only measure the initial linear region of the graph. At least three points should be used to draw the lines. Data points should be randomly on either side of the line. The data point positions should not be biased, with initial and final points below the line and middle points above the line, as observed in Fig. S9.

Response: We sincerely thank the reviewer for the helpful comments, patient explanation, and wonderful suggestions. To obtain more accurate values for the reactions with very low reaction rate, we have repeated many of the *in vitro* catalytic assays. And, based on the reviewer's suggestions, we have redrawn the related figures to match the y-axis with the maximal data.

2) The major insight coming from this work is the allowance of the mismatch being ligated. The authors propose that the viral active site is more flexible. When one considers flexibility in a structure, it is standardly considered that there are protein regions that adopt multiple conformations. Is this what the authors intended? Based on Figure S7, it appears that the DNA is flexible, with a lack of base pairing. A CT base should have two H-bonds, but it appears to have only one, suggesting strain. The position of the phosphodiester backbone appears to be maintained. Does the phosphodiester backbone overlay, when overlaying the protein chain for the CG and the CT complexes? If that is the case, are there more phosphodiester stabilizing residues in the *AsfV*Lig than in Lig1, 3, or 4? What is missing in the *AsfV*Lig from Lig 1, 3, and/or 4 that ensures W-C fidelity? Could *AsfV*Lig work on single stranded RNA or DNA?

Response: We sincerely thank the reviewer for the helpful comments. We agree with the reviewer that, with a lack of base pairing, CT could form two H-bonds. Fig. S7 has been renumbered as Fig. S6 in the revised manuscript. As depicted in Fig. S6D and the figure below, the electron density map clearly suggests that C:T only forms one H-bond in our *AsfV*LIG:DNA complex structures.

(A) Detailed conformation of the nick site C:T pair observed in the *AsfV*LIG:CT1 complex. The 2fo-fc electron density map is contoured at 1.5 sigma level. (B) *In vitro* ssDNA and ssRNA ligation assays catalyzed by WT *AsfV*LIG.

As suggested by reviewer 1, we have included one new panel in Fig. 5C in the revised manuscript, showing that the phosphodiester backbone, 5'-P, and 3'-OH are well overlaid in the *AsfvLIG:CG* and *AsfvLIG:CT1* complex. We have carefully compared our *AsfvLIG* structures with the *HsLIGs* structures, however, we did not identify any more residues that are missing in *AsfvLIG* and could form more stable interaction with phosphodiester and ensure W-C fidelity in *HsLIGs*.

Like all the characterized ATP-dependent DNA ligases, *AsfvLIG* catalyzes phosphodiester bond formation between adjacent 3'-OH and 5'-phosphate in DNA duplex through a similar three-step mechanism. However, in consistent with previous study (Showalter et al., 2006, Chemical Review, 106: 340-360), the obvious different DNA binding behavior, unique fold of the NTD domain, and unique residues at the nick site all confirm that *AsfvLIG* is one atypical DNA ligase. As revealed by our *in vitro* DNA binding and catalytic assays, *HsLig1* and *AsfvLIG* do not actively discriminate between match and mismatch during step 3; the discrimination or tolerance of mismatched DNAs solely occurs in step 2. We believe that the unique NTD domain and the unique residues at the nick site all contribute to the low fidelity of *AsfvLIG*. The extensive interactions between DNA and the NTD domain offer *AsfvLIG* with strong binding affinity to all matched and mismatched DNAs. In addition to matched DNAs, the strong binding also gives *AsfvLIG* enough opportunity to catalyze the ligation of certain mismatched DNAs. In incorporation with the NTD domain, the four nick site residues will help the reorientation of DNA 3'-OH and facilitate the ligation reaction. Compared to other nick site residues, Gln403 appears to be more important for the catalytic efficiency of *AsfvLIG*. As exemplified by the *AsfvLIG:CT2* structure (Fig. 5B), Gln403 can form H-bond interaction with the mismatched C:T pair at nick 3'-end; in addition to catalytic efficiency, such interaction definitely also contributes to the low fidelity of *AsfvLIG*.

Before any *AsfvLIG* structure was reported, Showalter and coworkers already demonstrated that *AsfvLIG* has very low adenylation activity towards DNAs with 3'-dideoxy- or 3'-amino-terminated nicks, compared to regular nick DNAs; these observations indicated that 3'-OH of the nick is a critical component of the active site architecture during 5'-P adenylation. The unique nick site residues of *AsfvLIG* could not be replaced by the corresponding residues of *HsLig1*, suggesting that they may work in an incorporative manner with other unique structure feature of *AsfvLIG*, especially NTD domain. The corporation between these unique structure features of *AsfvLIG* will help the reposition of the active site architecture and favor the nick sealing of matched and some mismatched DNAs.

As suggested by the reviewer, we did *in vitro* ssDNA and ssRNA ligation assay using the WT *AsfvLIG*. However, as depicted in the right panel of figure above, *AsfvLIG* could not support the ligation reaction of either ssDNA or ssRNA.

3) The authors included a catalytic reaction schematic, as requested. However, it is a simple one that could be used in a biochemistry paper without the structure. It would be more helpful and inclusive of the results being reported if that schematic includes the active site residues and geometry, as this is a structure paper where the active site

geometry is known.

Response: We sincerely thank the reviewer for the helpful comments. We totally agree with the reviewer that one real structure-based schematic is more informative than the one we presented. However, as we showed in the reaction schematic, the ligation reaction contains three consecutive steps. In this study, we are able to obtain one open-form ATP bound complex, two close-form *AsfvLIG*:DNA complex prior to reaction, and one close-form *AsfvLIG*:DNA complex after reaction. However, very unfortunate, several key intermediate structures (especially *AsfvLIG* with AMP linked with the catalytic Lys residue with or without nick DNA, close-form *AsfvLIG* structure in complex with 5'-pyrophosphate linked AppDNA) are still missing. Therefore, we are very sorry that we could not provide a structure-based schematic at present. The catalytic reaction schematic has been moved to Fig. 8A in the revised manuscript.

4) The conservation between AD and OB domains in porcine and human DNA ligases should be provided in text. It is interesting that despite the lower conservation, the structure is conserved.

Response: Thanks for the suggestion. The conservation between AD and OB domains in porcine and human DNA ligases have been described in the main text in the revised manuscript.

5) The reference to the Ligase IV structure was not added. Structural comparison to Ligase III and Ligase IV were not done, which could provide information on what makes the porcine virus enzyme unique. As above in #2, what is different from *AsfvLig* and the other Lig structures that allows for non-WC specificity? A part of this question is knowing what in the other lig structures enforces WC specificity.

Response: We sincerely thank the reviewer for the helpful comments and wonderful suggestions. The reference of *HsLIG4* (Reference 31) has been included in the revised manuscript. As suggested by both reviewers, we have done structural comparison between *AsfvLIG* and all the three *HsLIGs* (Figs. S4 and S9). The AD and OB domains are conserved in *AsfvLIG* and *HsLIGs*. Except the four unique nick site ligase residues in the AD and OB domains, we did not identify any more residues, which could form more stable interaction with the substrates or could affect the fidelity of the ligases.

The fidelity of the ligases are affected by many factors. Though both *HsLig1* and *HsLig3* are typical high fidelity DNA ligases, they attain fidelity through different strategies. As implicated by our *in vitro* DNA binding (Figs. 8D and S15) and catalysis (Figs. 7 and S11), *HsLig1* solely discriminates matched and mismatched DNAs during the reposition of active site architecture in step 2. However, in addition to active site architecture reposition, *HsLig3* can also attain part fidelity during step 3 (the nick closure step). Though there are a few *HsLig1*:DNA and *HsLig3*:DNA structures have been reported, more structures (especially the structures of *HsLig1* or *HsLig3* complexed with mismatched DNAs)

are required to fully understand their basis for discrimination between matched and mismatched DNAs.

In consistent with previous study (Showalter et al., 2006, Chemical Review, 106: 340-360), the obvious different DNA binding behavior, unique fold of the NTD domain, and unique residues at the nick site all confirm that *AsfvLIG* is one atypical DNA ligase. The unique NTD domain and the unique residues at the nick site all contribute to the low fidelity of *AsfvLIG*. The extensive interactions between DNA and the NTD domain offer *AsfvLIG* with strong binding affinity to all matched and mismatched DNAs. In addition to matched DNAs, the strong binding also gives *AsfvLIG* enough opportunity to catalyze the ligation of certain mismatched DNAs. In incorporation with the NTD domain, the four nick site residues will help the reorientation of DNA 3'-OH and facilitate the ligation reaction. Compared to other nick site residues, Gln403 appears to be more important for the catalytic efficiency of *AsfvLIG*. As exemplified by the *AsfvLIG*:CT2 structure (Fig. 5B), Gln403 can form H-bond interaction with the mismatched C:T pair at nick 3'-end; in addition to catalytic efficiency, such interaction definitely also contributes to the low fidelity of *AsfvLIG*.

6) Please include PDB ID for structures reported in this paper. It is helpful to the readers, if they wanted to look at them in a graphics program.

Response: Thanks for the helpful suggestion. The PDB codes for the four *AsfvLIG*:DNA complex structures have been provided in the "Accession Code" section in the main text and in the supplementary Table S6.

Reviewers' Comments:

Reviewer #1:

Remarks to the Author:

I thank the authors for the modifications that take into account almost all the remarks of the referees. In particular, I acknowledge the extensive additional experiments on a deletion mutant for the OB domain and on human HsLig1 and its mutants of the 4 residues interacting with the minor groove of the DNA substrate next to the nick site.

With these additional data and stressing the main results, the manuscript could have been transformed easily in a fully acceptable version. Instead, the authors introduced a lot of additional discussion which contains some problems.

First of all, the text added in order to address the referees points needs attentive proofreading and some language editing as it is generally of much poorer quality as the main manuscript.

Just a few examples :

L216: 2-fold.

L219: also investigated

L309: an open conformation

L366: efficiently

L367: A similar phenomenon ..., maybe due ...

L383: of the upstream pyrophosphate

L387-388: Though the overall fold ... This sentence is not clear and it certainly does not mean what the authors want to say.

L422: Very surprisingly, HsLig1

L426-430: incomprehensible phrases

L450: a cooperative manner

L452: Consistent with a previous study,

L458: The interaction

L568: "The exponential ... was fitted to the data" and not the other way round! Actually, the mathematical form of such progress curves is not necessarily exponential but can be more complex as detailed in the comments on Fig. 7.

A more serious point concerns the interpretation of the additional experiments, which do not point to any change of fidelity, only to reduced activity. The HsLig1 mutants behaves in this matter as the ones of AsFvLIG.

The author should have had the courage to state that the 4 studied residues are important for nick recognition and thus catalytic activity, but not have gone further. The question of the molecular basis for the fidelity or non-fidelity of the resealing reaction is still unresolved. This would have been a clear statement and would not have reduced at all the value of the present paper.

Through the lines, it becomes obvious that an important requirement for catalysis may be the preferential binding of the ligase at the nick site and probably here the 4 residues play their role. An increase in non-specific DNA-binding would lead to an inhibition of the enzyme by the regular stretches of dsDNA also present in the substrate.

The authors introduced a discussion whether the mismatch recognition / error tolerance takes place at step 2, adenylation of the nick phosphate, or at step 3, of the reaction, the resealing reaction.

Here supporting information is lacking. The adenylation of the nick phosphate can only be studied using radioactively marked ATP and autoradiography, but such experiments have not been out. The

reference for such studies is actually missing at line 415. I do not see how the presented results on electrophoretic mobility of ligated or unligated DNA oligos could yield an information about the possible adenylation and the built-up of adenylated intermediates.

I have another issue which is rather an observation and not a direct issue:

In Fig. 7A, the fit of an exponential function has been used in order to describe the product yield in function of the time. This may be a suitable approach, but only under certain conditions for k_{cat} and K_M . In general, the product build-up is described by an integrated Michaelis-Menton equation (see Larsson, G., Nyman, P. O., and Kvassman, J. O. (1996) *J. Biol. Chem.* 271, 24010–24016.) I have been intrigued by the fact that the relatively poor activities for mismatched base pairs rapidly level off. This could actually be due to a strong inhibitory effect of the product in the case of a mismatched base at the ligation site. The ligase may still sense the perturbation of the DNA structure at the position of the mismatch and stay bound to the DNA leading to a product inhibition.

Fig. 8B is difficult to understand, it should be described more efficiently what is shown. I suppose the hSLIG1 structure contains an adenylated DNA.

The presentation of Fig. 8C is another problem because of the normalisation of the protein-bound DNA fraction to one at a protein concentration of $1.6 \mu\text{M}$. From Fig. 8C, I would estimate a K_D of about $1 \mu\text{M}$ for the binding of the ligases to the different DNA substrates. At about 1.6 times the K_D you are still far from saturation and the fraction of protein-bound DNA would rather correspond to 60 % than to 100 %. It would be better not to normalize and to use the result of the quantification as it is.

[Following is a figure showing the theoretical dependence of the protein-bound DNA fraction in function of the protein concentration for a K_D of $1 \mu\text{M}$. See pdf file of the review]

We sincerely thank the reviewer for reading our manuscript with great care, and for all the helpful comments, encouragement, and criticisms. Based on these comments and suggestions, we have carefully revised our manuscript with major changes highlighted in red. The following are our point-to-point responses to the reviewer's comments.

- 1) I thank the authors for the modifications that take into account almost all the remarks of the referees. In particular, I acknowledge the extensive additional experiments on a deletion mutant for the OB domain and on human HsLig1 and its mutants of the 4 residues interacting with the minor groove of the DNA substrate next to the nick site.

Response: We sincerely thank the reviewer for all the nice comments.

- 2) With these additional data and stressing the main results, the manuscript could have been transformed easily in a fully acceptable version. Instead, the authors introduced a lot of additional discussion which contains some problems.

Response: We sincerely thank the reviewer for the encouragement and criticisms as well. Based on the reviewer's comments, we have carefully revised the discussion section in the revised manuscript.

- 3) First of all, the text added in order to address the referees points needs attentive proofreading and some language editing as it is generally of much poorer quality as the main manuscript.

Response: We sincerely thank the reviewer for the helpful comments and criticisms. Our manuscript has been edited by a language service company.

Just a few examples :

- 4) L216: 2-fold.

Response: Thanks for the correction. The mistake has been fixed in the revised manuscript.

- 5) L219: also investigated

Response: Thanks for the correction. The mistake has been fixed in the revised manuscript.

- 6) L309: an open conformation

Response: Thanks for the suggestion. We have replaced "open form conformation" with "an open conformation" in the revised manuscript.

7) L366: efficiently

Response: Thanks for the correction. The mistake has been fixed in the revised manuscript.

8) L367: A similar phenomenon ..., maybe due ...

Response: Thanks for the correction. The mistake has been fixed in the revised manuscript.

9) L383: of the upstream pyrophosphate

Response: Thanks for the suggestion. The original sentence has been replaced by “In the third step, the 3’-OH of the upstream DNA attacks the pyrophosphate group of AppDNA,” in the revised manuscript.

10) L387-388: Though the overall fold ... This sentence is not clear and it certainly does not mean what the authors want to say.

Response: Thanks for the helpful comment. The sentence has been replaced by “Though the domain arrangement of the non-catalytic form” in the revised manuscript.

11) L422: Very surprisingly, HsLig1

Response: Thanks for the correction. The mistake has been fixed in the revised manuscript.

12) L426-430: incomprehensible phrases

Response: Thanks for the helpful comments. These sentences have been replaced by “Based on the HsLig1:DNA complex structure (PDB_ID: 1X9N), it has been previously proposed that HsLig1 can cause some local distortion on the duplex; this distortion is important for the alignments of the 3’-OH and the adenylated 5’-P for nick sealing” in the revised manuscript.

13) L450: a cooperative manner

Response: Thanks for the correction. We have replaced “an incorporative manner” with “a cooperative manner” in the revised manuscript.

14) L452: Consistent with a previous study,

Response: Thanks for the suggestion. We have replaced “In consistent with previous study” with “Consistent with a previous study” in the revised manuscript.

15) L458: The interaction

Response: Thanks for the suggestion. We have replaced “The incorporation” with “The interaction” in the revised manuscript.

16) L568: “The exponential ... was fitted to the data” and not the other way round! Actually, the mathematical form of such progress curves is not necessarily exponential but can be more complex as detailed in the comments on Fig. 7.

Response: We sincerely thank the reviewer for the helpful comments. Please see our detailed response in question #20.

17) A more serious point concerns the interpretation of the additional experiments, which do not point to any change of fidelity, only to reduced activity. The HsLig1 mutants behave in this matter as the ones of AsfvLIG. *The author should have had the courage to state that the 4 studied residues are important for nick recognition and thus catalytic activity, but not have gone further. The question of the molecular basis for the fidelity or non-fidelity of the resealing reaction is still unresolved. This would have been a clear statement and would not have reduced at all the value of the present paper.*

Response: We sincerely thank the reviewer for the helpful comments and suggestions. Though mutation of the nick site residues significantly lowered the ligation activities of AsfvLIG and HsLIG1, we agree with the reviewer that the molecular basis underlying the fidelity or non-fidelity of the two ligases is still unresolved. We thank the reviewer for confirming the functional importance of these nick site residues. All these points have been reflected in the revised manuscript. Also, based on the reviewer’s suggestions, we have deleted all the fidelity-related discussion in the revised manuscript.

18) Through the lines, it becomes obvious that an important requirement for catalysis may be the preferential binding of the ligase at the nick site and probably here the 4 residues play their role. An increase in non-specific DNA-binding would lead to an inhibition of the enzyme by the regular stretches of dsDNA also present in the substrate.

Response: We sincerely thank the reviewer for the very helpful and thoughtful comments. We have mentioned the potential inhibitory effects of the DNAs in the revised manuscript.

19) The authors introduced a discussion whether the mismatch recognition / error tolerance takes place at step 2, adenylation of the nick phosphate, or at step 3, of the reaction, the resealing reaction. Here supporting information is lacking. The adenylation of the nick phosphate can only be studied using radioactively marked ATP and autoradiography, but such

experiments have not been out. The reference for such studies is actually missing at line 415. I do not see how the presented results on electrophoretic mobility of ligated or unligated DNA oligos could yield an information about the possible adenylation and the built-up of adenylated intermediates.

Response: We are so sorry for the inaccurate interpretation of the ligation results. We sincerely thank the reviewer for pointing out that radioactively marked ATP and autoradiography are required to visualize the adenylation of the nick phosphate. Very unfortunately, our lab does not have the license to perform the assay using radioactive materials. Therefore, as suggested by the reviewer, we mainly focus on the activity of *AsfV*LIG and *Hs*LIG1 in the revised manuscript. The discussion related to the step 2 and step 3 of the ligation reaction has been deleted. The reference associated with the step 2 was not included, owing to the removal of the discussion.

20) I have another issue which is rather an observation and not a direct issue: In Fig. 7A, the fit of an exponential function has been used in order to describe the product yield in function of the time. This may be a suitable approach, but only under certain conditions for k_{cat} and K_M . In general, the product build-up is described by an integrated Michaelis-Menton equation (see Larsson, G., Nyman, P. O., and Kvassman, J. O. (1996) *J. Biol. Chem.* 271, 24010–24016.) I have been intrigued by the fact that the relatively poor activities for mismatched base pairs rapidly level off. This could actually be due to a strong inhibitory effect of the product in the case of a mismatched base at the ligation site. The ligase may still sense the perturbation of the DNA structure at the position of the mismatch and stay bound to the DNA leading to a product inhibition.

Response: We sincerely thank the reviewer for all the helpful and thoughtful comments. We totally agree with the reviewer that Michaelis-Menton equation might be a more general equation for k_{cat} and K_M calculation. We are so sorry that we could not run the ligation assays using the radioactive substrates, which will produce more reliable data for Michaelis-Menton equation. The exponential function we used maybe is not as accurate as the Michaelis-Menton equation, it allows us to get the apparent K_{obs} . As supported by the gel analysis, we believe that our conclusions should be correct.

We thank the reviewer for pointing out the possibility that *Hs*LIG1 could sense the perturbation of the DNA structure at the position of the mismatch, which leads to a product inhibition. We are working on the crystallographic study of *Hs*LIG1. Hopefully, we could solve some structures of *Hs*LIG1 in complex with mismatch DNAs, which will provide

some structural insights into the fidelity and activity of HsLIG1. In addition to structural study, dynamic study of *HsLIG1* is also important for understanding the inhibitory effects of the mismatched products. Without these results in hand, we could not discuss this phenomenon in the manuscript.

21) Fig. 8B is difficult to understand, it should be described more efficiently what is shown. I suppose the *hsLIG1* structure contains an adenylated DNA. The presentation of Fig. 8C is another problem because of the normalisation of the protein-bound DNA fraction to one at a protein concentration of 1.6 μM . From Fig. 8C, I would estimate a K_D of about 1 μM for the binding of the ligases to the different DNA substrates. At about 1.6 times the K_D you are still far from saturation and the fraction of protein-bound DNA would rather correspond to 60 % than to 100 %. It would be better not to normalize and to use the result of the quantification as it is.

Response: We sincerely thank the reviewer for the helpful comments. In Fig. 8B, the non-catalytic *AsfvLIG*:DNA complex structure is compared with the *HsLIG1* structure in complex with adenylated DNA. The legend of Fig. 8B has been updated in the revised manuscript.

We also thank the reviewer for capturing our mistake on Fig. 8C. As depicted in Fig. S15, nearly all DNAs were bound by *HsLIG1* at a concentration of 1.6 μM . We did not normalize the DNA fraction bound at a protein concentration of 1.6 μM , but we forgot to show the error-bars previously. As suggested by the reviewer, the direct quantification results have been used in the updated figure.

Reviewers' Comments:

Reviewer #1:

Remarks to the Author:

I am very grateful to the authors for taking into account the comments of my last review and I am now in favor of a prompt publication of the manuscript.

We sincerely thank both reviewers for reading our manuscript with great care. We would also like to thank the reviewers for all their previous and current comments and suggestions, which have significantly improved the quality of our manuscript. The following are our point-to-point responses to the reviewers' comments.

REVIEWERS' COMMENTS:

Reviewer #1 (Remarks to the Author):

I am very grateful to the authors for taking into account the comments of my last review and I am now in favor of a prompt publication of the manuscript.

Wim P. Burmeister

Wim P. Burmeister

Response: We sincerely thank Prof. Wim P. Burmeister for all the good comments and encouragements.